# HGF/R-spondin1 rescues liver dysfunction through the induction of Lgr5+ liver stem cells

Yuan Lin[1,2,3], Zhe-Ping Fang[4], Hong-Juan Liu[5], Li-Jing Wang[6], Zhiqiang Cheng[7], Na Tang[7], Tingting Li[1,2,3], Tengfei Liu[1,2,3], Hai-Xiong Han[4], Guangwen Cao[8], Li Liang[1,2,3], Yan-Qing Ding[1,2,3] & Wei-Jie Zhou[1,2,3]

Induction of endogenous adult stem cells by administering soluble molecules provides an advantageous approach for tissue damage repair, which could be a clinically applicable and cost-effective alternative to transplantation of embryonic or pluripotent stem cell-derived tissues for the treatment of acute organ failures. Here, we show that HGF/Rspo1 induce liver stem cells and rescue liver dysfunction. Carbon tetrachloride treatment promotes both fibrosis and Lgr5+ liver stem cell proliferation, whereas Lgr5 knockdown worsens fibrosis. Injection of HGF in combination with Rspo1 increases the number of Lgr5+ liver stem cells and improves liver function by attenuating fibrosis. We observe Lgr5+ liver stem cells in human liver fibrosis tissues, and once they are isolated, these cells are able to form organoids, and treatment with HGF/Rspo1 promotes their expansion. We suggest that Lgr5+ liver stem cells represent a valuable target for liver damage treatment, and that HGF/Rspo1 can be used to promote liver stem cell expansion.

[1] Department of Pathology, School of Basic Medical Sciences, Southern Medical University, Guangzhou, Guangdong 510515, China. [2] Department of Pathology, Nanfang Hospital, Southern Medical University/The First School of Clinical Medicine, Southern Medical University, Guangzhou, Guangdong 510515, China. [3] Guangdong Provincial Key Laboratory of Molecular Oncologic Pathology, Guangzhou, Guangdong 510515, China. [4] Department of Hepatobiliary Surgery, Taizhou Hospital of Zhejiang Province, Wenzhou Medical University, Linhai, Zhejiang 317000, China. [5] Department of Bioinformatics, School of Basic Medical Sciences, Southern Medical University, Guangzhou, Guangdong 510515, China. [6] Vascular Biology Research Institute, Basic Course of School, Guangdong Pharmaceutical University, Guangzhou 510006, China. [7] Department of Pathology, Shenzhen People's Hospital, Shenzhen, Guangdong 515020, China. [8] Department of Epidemiology, Second Military Medical University, Shanghai 200433, China. Yuan Lin and Zhe-Ping Fang contributed equally to this work. Li Liang, Yan-Qing Ding and Wei-Jie Zhou jointly supervised this work. Correspondence and requests for materials should be addressed to L.L. (email: lli@smu.edu.cn) or to Y.-Q.D. (email: dyqgz@126.com) or to W.-J.Z. (email: weijiezhouum@163.com)

The liver is a vital organ of the digestive system in vertebrates. It has a wide range of functions, including detoxification, the synthesis of critical plasma proteins such as albumin, and the production of biochemicals that are necessary for digestion. As a result of these diverse and vital functions, loss of liver function results in organ failure and subsequent hypotension, hypoglycemia, encephalopathy, and death within days[1,2]. Currently, there is no way to compensate for long-term loss of liver function, although new liver dialysis techniques can be used in the short term.

Leucine-rich repeat-containing G-protein-coupled receptor 5 (Lgr5) is a stem cell marker in various organs/tissues including the gut, stomach, hair follicle, mammary gland, kidney, and ovary[3–7], but it is not expressed in the homeostatic adult liver[8]. However, following both CCl4-induced acute damage and "oval cell response" damage (3,5-diethoxycarbonyl-1,4-dihydrocollidine and CDE), a population of Lgr5+ stem cells/progenitors has recently been demonstrated to actively contribute to liver regeneration via de novo generation of hepatocytes and ductal cells[8].

Roof plate-specific spondin 1 (R-spondin1, Rspo1), a secreted ~35-kDa molecule, synergizes with soluble Wnt3a to induce LRP6 phosphorylation and to promote the nuclear accumulation of β-catenin for cellular proliferation, differentiation, and stem cell maintenance[9–13]. The administration of recombinant or adenoviral Rspo1 alleviates intestinal injury and oral mucositis induced by chemoradiotherapy[14–16], experimental colitis[17], and systemic graft-vs.-host disease (GVHD)[18]. Rspo1 is absolutely required for in vitro culture of intestinal organoids and liver organoids derived from Lgr5+ stem cells[8,19,20]. Mechanistically, Rspo1 binds to the leucine-rich repeat containing G-protein-coupled receptor 4–6 (Lgr4–6) with a high affinity and enhances Lrp6 phosphorylation at $S^{1490}$ ($pS^{1490}$Lrp6) by direct inhibition of two membrane-bound E3 ligases (RNF43 and ZNRF3) for ubiquitination and degradation of Wnt receptors[9–12]. However, the roles of Rspo1 in the liver are still unknown.

Our previous studies demonstrated that the systemic administration of a combination of recombinant Rspo1 and Slit2 preventively reduces Lgr5+ intestine stem cell loss, mitigates gut injury, and protects mice from death caused by lethal doses of chemoradiation, without concomitantly decreasing tumor sensitivity to chemotherapy[16]. In this study, we reported that the deletion of Lgr5+ liver stem cells increased CCL4-induced liver fibrosis, whereas a combination of HGF and Rspo1 significantly induced more Lgr5+ liver stem cells, thereby facilitating the recovery from liver dysfunction and attenuating liver fibrosis induced by CCl4. Moreover, Lgr5+ liver stem cells were isolated from human liver fibrosis tissues, and single human Lgr5+ liver cells expanded as organoids, which could be further enhanced by treatment with HGF and Rspo1 proteins in vitro. These findings indicate that Lgr5+ liver stem cells are crucial for recovery from liver dysfunction, and that the combination of HGF and Rspo1, which induces Lgr5+ stem cells, might be able to be used for liver fibrosis therapy.

## Results

**Lgr5+ stem cells were induced during liver fibrosis process.** The induction of Lgr5+ liver stem cells by a single dose of CCL4 was recently reported[8]. We reproduced these results and observed that the Lgr5 expression induced by a single dose of CCL4 was reduced following liver recovery (Supplementary Fig. 1). Next, we detected whether Lgr5+ liver stem cells were maintained upon chronic damage during CCL4-induced liver fibrosis development. We used Lgr5-enhanced green fluorescent protein (eGFP)–internal ribosome entry site (IRES)–CreERT2 (Lgr5-GFP)

mice[6] to detect the expression of Lgr5-GFP in the liver. Eight-week-old Lgr5-GFP mice were intraperitoneally (i.p.) injected with CCl4 (diluted at a ratio of 1:4 in olive oil) or olive oil alone (2 ml/kg body weight) twice a week for 6 weeks (Supplementary Fig. 2a). In oil-treated control Lgr5-GFP mice, Lgr5-GFP was essentially undetectable. Upon CCl4 treatment, clear GFP-positive cells were observed from day 1 to day 40 (Supplementary Fig. 2b). The expression of Lgr5 was confirmed using qRT-PCR assay, which demonstrated an ~2–3-fold increased induction of Lgr5 in CCl4-treated mice liver compared with oil-treated mice liver (Supplementary Fig. 2c). Lgr5 expression peaked at day 5 and maintained this level during the development of liver fibrosis. These Lgr5+ cells expressed Sox9, a relatively broad ductal progenitor marker (Supplementary Fig. 3a), but did not express mature hepatocyte cell markers such as Hnf4a (Supplementary Fig. 3b).

Next, we investigated whether Lgr5+ cells induced upon chronic damage are liver stem cells. Single Lgr5-GFP+ cells were sorted on day 40, from Lgr5-GFP mice continuously treated with CCl4, as described in Supplementary Fig. 2a. Sorted cells, cultured in stem cells medium, rapidly divided and formed organoid structures that were maintained by weekly passaging (Supplementary Fig. 4a). Lgr5+ cells sorted from the liver fibrosis model formed organoids, which were similar in number and size to those formed by cells sorted from the 1XCCL4 damage model (Supplementary Fig. 4b, c). Moreover, when the Lgr5+ cells sorted from the liver fibrosis model were cultured in a differentiation medium (DM), they expressed mature hepatic genes (Supplementary Fig. 4d), and abundant amounts of albumin and AAT were secreted into the medium (Supplementary Fig. 4e, f). The differentiated cells were competent for accumulated glycogen (Supplementary Fig. 4g) and low-density lipoprotein (LDL) uptake (Supplementary Fig. 4h). These results suggest that these Lgr5+ cells that are induced upon chronic damage are liver stem cells.

**Transplantation of Lgr5+ cells attenuated liver fibrosis.** We next asked whether Lgr5+ liver stem cells supported the recovery from acute damage or chronic damage. Using FACS, we isolated Lgr5-GFP+ liver stem cells from Lgr5-GFP mice treated with 1XCCL4, and injected these cells intrasplenically into the wild-type C57 mice with acute liver damage (single CCL4 treatment) or chronic liver damage (liver fibrosis model, 2XCCL4 treatment/week for 6 weeks, Fig. 1a). We used mouse primary hepatocytes (PH) as controls. GFP-positive cells were detected in mice transplanted with Lgr5-GFP+ liver stem cells on day 40, but not in mice transplanted with PH (Fig. 1b). These Lgr5-GFP+ cells co-stained with a ductal progenitor marker Sox9 (Supplementary Fig. 5). To our surprise, in the acute liver damage model, both Lgr5-GFP+ liver stem cells and PH-treated mice demonstrated normal serum ALT and AST (Supplementary Fig. 6). In the chronic liver damage model, the mice exhibited attenuated fibrosis phenotypes when transplanted with Lgr5-GFP+ liver stem cells but not PH (Fig. 1c–e). The therapeutic effect was dose dependent. Transplantation of $10^5$ Lgr5+ liver stem cell transplantation reduced the fibrotic area and significantly decreased the serum ALT and AST levels in CCL4-induced mice (Fig. 1c–e). However, because the lineage-tracing model is not currently available in our lab, it is not clear to which cell types do these Lgr5+ cells contribute to after transplantation. According to an in vitro differentiation assay, transplanted Lgr5+ liver stem cells might primarily generate more hepatocytes in the host. Together, these results provided evidence for the therapeutic effect of Lgr5-GFP+ liver stem cells in the treatment of liver injuries, especially chronic liver fibrosis.

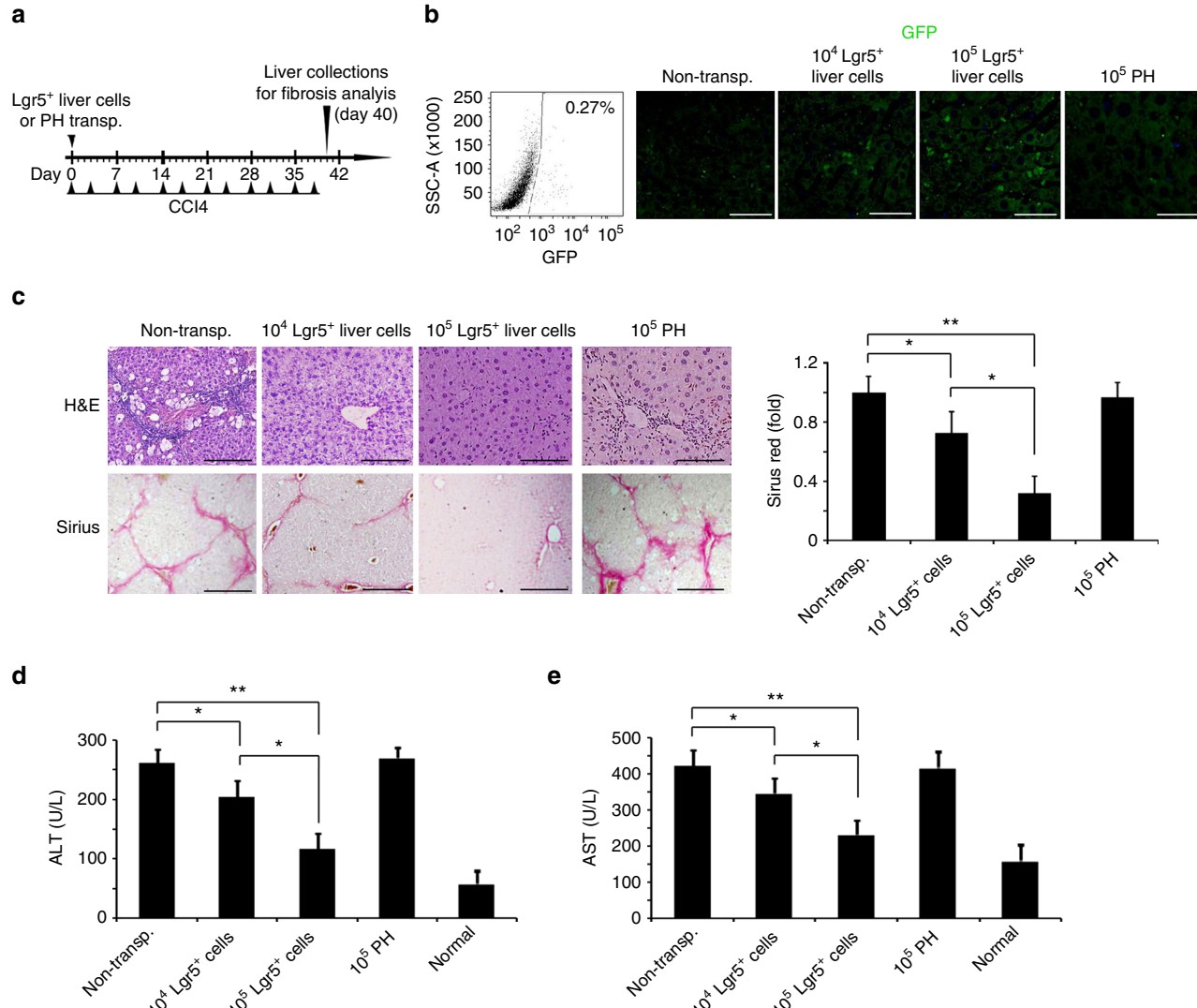

**Fig. 1** Lgr5[+] liver stem cells transplantation decreased liver fibrosis. **a** Schematic overview of the experimental setup. Eight-week-old wild-type C57 mice were i.p. injected with CCL4 (2 ml/kg, Sigma-Aldrich) dissolved in olive oil at a ratio of 1:4, or olive oil alone (2 ml/kg) twice a week for 6 weeks. Lgr5-GFP[+] liver stem cells or primary hepatocyte (PH) derived from Lgr5-GFP mice were transplanted into the liver fibrosis mice by intrasplenical injection on day 0. **b** Lgr5[+] cells were isolated from Lgr5-GFP mice treated with CCL4 by FACS assay for transplantation (left). The liver Lgr5 expression was stained using anti-GFP antibody in mice with Lgr5[+] cells or PH transplantation (right) on day 40. $n = 10$ mice, scale bars, 200 μm. **c, d** Lgr5[+] liver stem cells transplantation decreased CCL4-induced liver fibrosis and recovered liver functions. Lgr5-GFP[+] liver stem cells or PH transplantation were described in **a**; the livers were harvested and stained using H&E and Sirius Red for fibrosis analysis and quantification of positive-staining areas measured by Image J software (**c**). The serum was harvested for ALT and AST analysis (**d, e**). For **c**, scale bars, 200 μm, the results are shown as mean ± s.d. of five independent sections taken randomly per mice and a total of 50 tissue specimens in each group ($n = 10$ mice) *$p < 0.05$, **$p < 0.01$. For **d, e**, triplicates for each condition were analyzed. The results are shown as mean ± s.d. of three independent experiments. *$p < 0.05$, **$p < 0.01$

**Knockdown of Lgr5 aggravated liver fibrosis caused by CCl4.** We next investigated whether endogenous Lgr5[+] liver stem cells played a crucial role in the development of liver fibrosis. To answer this question, we designed a tool to delete or disable Lgr5[+] liver stem cells. We then investigated whether Lgr5 expression was essential for the stem cell properties of Lgr5[+] liver stem cells. In the intestine, using a conditional mouse model, de Lau et al.[9] demonstrated that *Lgr5* deletion does not result in intestinal disruption, unless it is concomitantly deleted with the homolog *Lgr4*. However, although intestinal Lgr5[+] cells expressed both Lgr5 and Lgr4 proteins, liver Lgr5[+] cells expressed Lgr5 alone (Supplementary Fig. 7a, b). Consistently, for intestinal Lgr5[+] cells, we had to knock down both *Lgr5* and *Lgr4* to inhibit their organoids formation ability (Supplementary Fig. 7c–e), whereas

for liver Lgr5[+] cells, we abrogated their organoids formation ability by *Lgr5* knockdown alone (Fig. 2a, b; Supplementary Fig. 8). Single Lgr5-GFP[+] cells were sorted from CCL4-treated *Lgr5-GFP* mice at day 5 after CCL4 treatment and then infected with adenoviral *Lgr5* shRNA (Ad-*Lgr5* shRNA); both the Lgr5 protein expression and Lgr5 messenger RNA (mRNA) expression were decreased at 72 h after adenovirus infection (Fig. 2a; Supplementary Fig. 8). The sorted cells were cultured as previously described[8]. We observed that the knockdown of *Lgr5* inhibited the in vitro formation of liver organoids (Fig. 2b; Supplementary Fig. 9). These results indicated that without Lgr5 expression, Lgr5[+] liver stem cells lost their stem cell properties. Moreover, the results indicated that Ad-*Lgr5* shRNA can be used to mimic Lgr5[+] liver stem cell loss of function for further study.

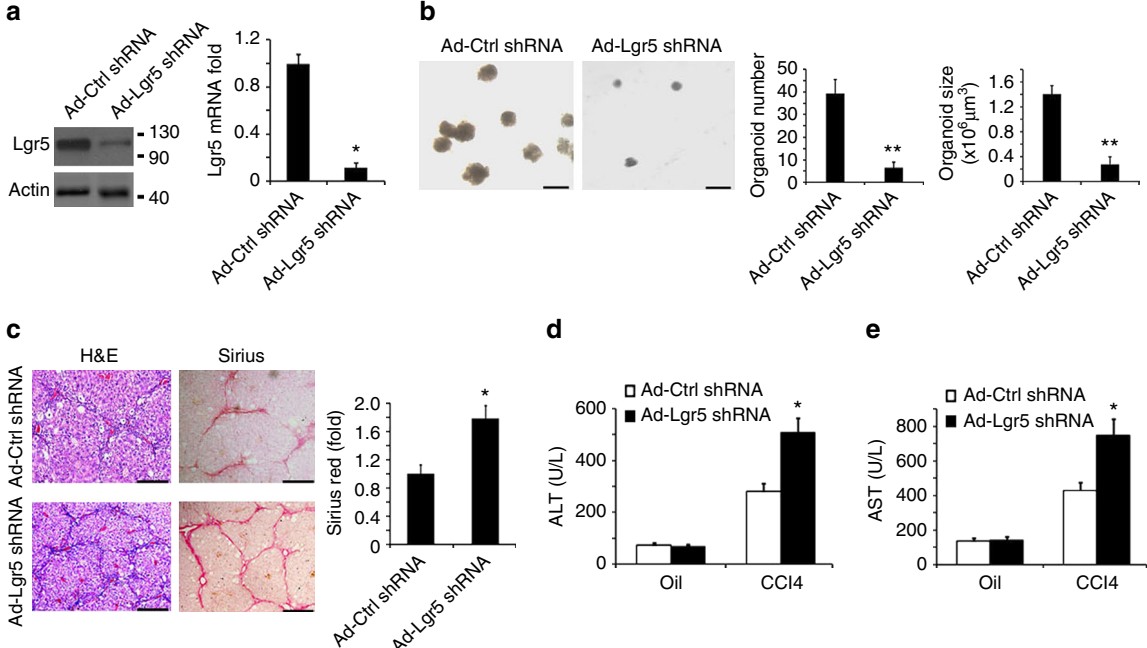

**Fig. 2** Knocking down of Lgr5 expression decreases Lgr5[+] cells' stem cell properties and increases liver fibrosis. **a** Lgr5 was knocked down by Ad-Lgr5 shRNA. Lgr5[+] liver stem cells were isolated and infected with Ad-Lgr5 shRNA, or nonsense control shRNA Ad-Ctrl shRNA, western blot, and qRT-PCR were used to analyze the proteins level (left) and the mRNA level (right) of Lgr5. Actin was used as a loading control. **b** Lgr5[+] liver stem cells failed to form an organoid when Lgr5 was downregulated by Ad-Lgr5 shRNA. Lgr5[+] liver stem cells were isolated and infected with Ad-Lgr5 shRNA or Ad-Ctrl shRNA, and cultured in stem cells medium in Matrigel, and the number and sizes of the liver organoid were measured at day 14. **b–e** Ad-Lgr5 shRNA increased CCL4-induced liver fibrosis. As shown in Supplementary Fig. 10a, Lgr5-GFP mice infected with Ad-Lgr5 shRNA or Ad-Ctrl shRNA were administered with CCL4 (2 ml/kg i.p.) twice/week for 6 weeks, respectively. The representative histology of H&E and Sirius Red, and the quantification of positive-staining areas is measured by Image J software (**c**). Serum levels of ALT and AST were measured (**d**, **e**). For **a**, **d**, **e**, triplicates for each condition were analyzed. The results are shown as mean ± s.d. of three independent experiments. *$p < 0.05$. For **b**, the results are shown as mean ± s.d. of three independent experiments. **$p < 0.01$. For **c**, scale bars, 200 μm, the results are shown as mean ± s.d. of five independent sections taken randomly per mice and a total of 50 tissue specimens in each group ($n = 10$ mice), *$p < 0.05$

Next, we infected the mice with Ad-*Lgr5* shRNA to disable endogenous Lgr5[+] liver stem cells to investigate whether Lgr5[+] liver stem cells played a crucial role in liver fibrosis development. We used a CCL4-induced mouse model of liver fibrosis. Eight-week-old C57 mice were injected with CCl4 twice a week for 6 weeks, and Ad-*Lgr5* shRNA or nonsense control shRNA adenovirus (Ad-Ctrl shRNA) was injected into the tail vein of mice from day 0 and once a week until the mice were killed (Supplementary Fig. 10a). We observed that Ad-*Lgr5* shRNA infection decreased the mRNA level of Lgr5 (Supplementary Fig. 10b), and there were no Lgr5[+] cells induced by CCl4 when mice were treated with Ad-*Lgr5* shRNA (Supplementary Fig. 10c, d). Surprisingly, Ad-*Lgr5* shRNA-treated mice exhibited more severe liver injury and fibrosis than Ad-Ctrl shRNA-treated mice, as detected by H&E staining and Sirius staining (Fig. 2c). Furthermore, the serum levels of ALT and AST (Fig. 2d, e) were significantly increased in Ad-Lgr5 shRNA-treated mice compared with those observed in the Ad-Ctrl shRNA-treated mice.

To further evaluate the function of Lgr5 in liver Lgr5[+] stem cells, we generated *Lgr5* knockout adenovirus using a CRISPR/Cas9 system. The mRNA of *Cas9* and guide RNA (gRNA), which targets the first exon of the *Lgr5* gene, were packaged into an adenovirus (Supplementary Fig. 11a). Single Lgr5[+] cells were sorted from CCL4-treated *Lgr5-GFP* mice at day 5 after CCL4 treatment, and then infected with the adenovirus encoding *Cas9* and *Lgr5* gRNA (Ad-*Lgr5* gRNA) or control nonsense gRNA (Ad-Ctrl gRNA); the Lgr5 protein and mRNA expression were decreased at 72 h after adenovirus infection (Supplementary

Fig. 11b). The sorted cells were cultured as previously described[8]. We observed that the knockout of *Lgr5* inhibited the in vitro formation of liver organoids (Supplementary Fig. 11c). We then infected the mice with adenovirus encoding *Cas9* and *Lgr5* gRNA to disable the Lgr5[+] liver stem cells in vivo. We used a CCL4-induced mouse model of liver fibrosis. Eight-week-old C57 mice were injected with CCl4 twice a week for 6 weeks. Adenovirus encoding *Cas9* and *Lgr5* gRNA (Ad-Lgr5 gRNA) or control nonsense gRNA (Ad-Ctrl gRNA) were injected into the tail vein of mice from day 0 and once a week until the mice were killed. We observed that infection with adenovirus encoding *Cas9* and *Lgr5* gRNA infection decreased the protein and mRNA levels of Lgr5 in the liver (Supplementary Fig. 11d). Surprisingly, mice treated with adenovirus encoding *Cas9* and *Lgr5* gRNA-treated mice exhibited more severe liver injury and fibrosis than mice treated with adenovirus encoding *Cas9* and control gRNA-treated mice, as detected by H&E staining and Sirius staining (Supplementary Fig. 11e). Furthermore, the serum levels of ALT and AST (Supplementary Fig. 11f, g) were significantly increased in mice treated with adenovirus encoding *Cas9* and *Lgr5* gRNA-treated mice compared with mice treated with adenovirus encoding *Cas9* and control gRNA-treated mice. To confirm the potency of CRISPR/Cas9 targeting *Lgr5*, we detected the Lgr5 expression in the intestine in mice treated with adenovirus encoding *Cas9* and *Lgr5* gRNA. We observed that infection with adenovirus encoding *Cas9* and *Lgr5* gRNA decreased the mRNA level of Lgr5 (Supplementary Fig. 12a) and the number of Lgr5[+] cells in the intestine (Supplementary Fig. 12b).

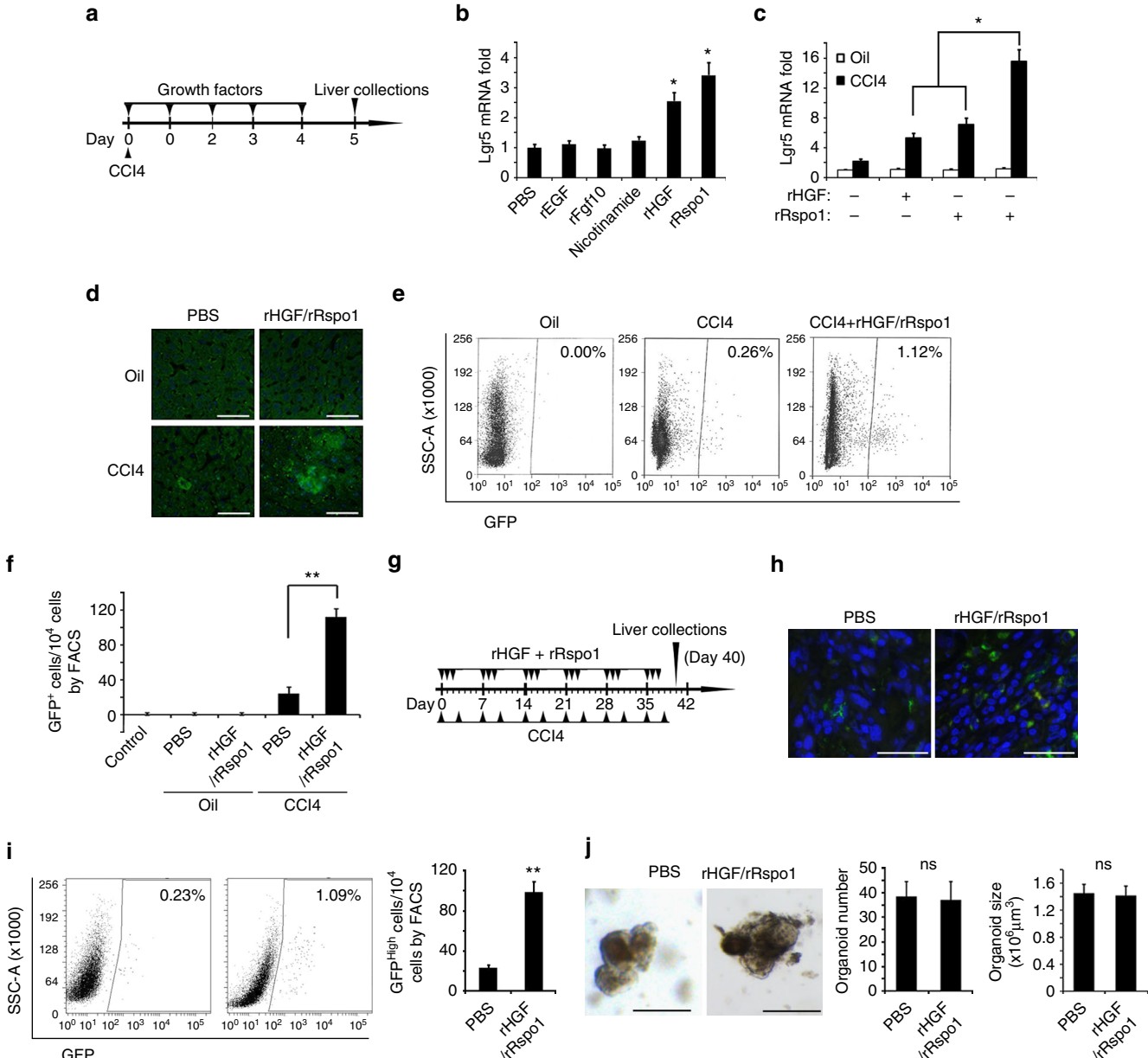

**Fig. 3** rHGF/rRspo1 induces Lgr5[+] liver stem cells upon CCL4 damage. **a** Schematic outline upon 1XCCL4 damage for **b**–**f**. **b**, **c** The Lgr5 expression was measured by qRT-PCR analysis using Lgr5 primer. Eight-week-old C57BL/J mice were injected with CCL4 and different growth factors (0.1 μg/mice/time), as indicated in **a**, and the livers were harvested and analyzed. **d**–**f** Eight-week-old Lgr5-GFP mice were injected with CCL4 and rHGF or/and rRspo1 proteins (0.1 μg/mice/time), as indicated in **a**, Lgr5 expression was measured using an immunofluorescent assay with anti-GFP antibody (**d**), Lgr5[+] cell numbers were analyzed by FACS assay **e**, **f**. **g**. Schematic outline upon chronic CCL4 damage for **h**–**j**. **h**, **i** Eight-week-old Lgr5-GFP mice were injected with CCL4 and rHGF or/and rRspo1 proteins (0.1 μg/mice/time), as indicated in **g**, and Lgr5 expression was measured using an immunofluorescent assay with anti-GFP antibody on day 40 (**h**), Lgr5[+] cell numbers were analyzed by FACS assay **i**. **j**, single Lgr5[+] liver stem cells were isolated from liver fibrosis model with rHGF plus rRspo1 treatment described as **g** on day 40 (rHGF/rRspo1) or from 1XCCl4-injured liver treated with PBS on day 5 (PBS), which was cultured to grow into organoids. The represented organoid pictures were shown, and the organoids number and organoids size were analyzed. For **b** and **c**, $n = 5$ mice/group. Scale bars, 200 μm (**d**, **g**). For **b**, **c**, triplicates for each condition were analyzed. The results are shown as mean ± s.d. of three independent experiments. *$p < 0.05$. For **f**, **i**, **j**, the results are shown as mean ± s.d. of three independent experiments. **$p < 0.01$

These results suggested that the knockdown of Lgr5 aggravated CCL4-induced liver fibrosis. Together, our results suggest that Lgr5[+] liver stem cells may work to alleviate Lgr5 expression-dependent liver fibrosis.

**HGF/Rspo1 induced Lgr5[+] liver stem cells upon CCl4 treatment.** We next explored how to induce more endogenous Lgr5[+] liver stem cells in vivo. Our previous studies demonstrated that

the in vitro intestine organoid culture factors, Rspo1 and Slit2, induced Lgr5[+] intestine stem cells in vivo[16]. Therefore, we then tested whether the growth factors involved in the Lgr5[+] liver stem cells culture medium in vitro induced the Lgr5[+] liver stem cells in vivo. For this purpose, recombinant His-tagged EGF, Fgf10, HGF, and Rspo1 (rEGF, rFgf10, rHGF, and rRspo1) were produced (Supplementary Fig. 13). Eight-week-old *Lgr5-GFP* mice were injected with a single dose of CCL4 at day 0, the growth factors (0.1 μg per mice and once per day) were injected into the

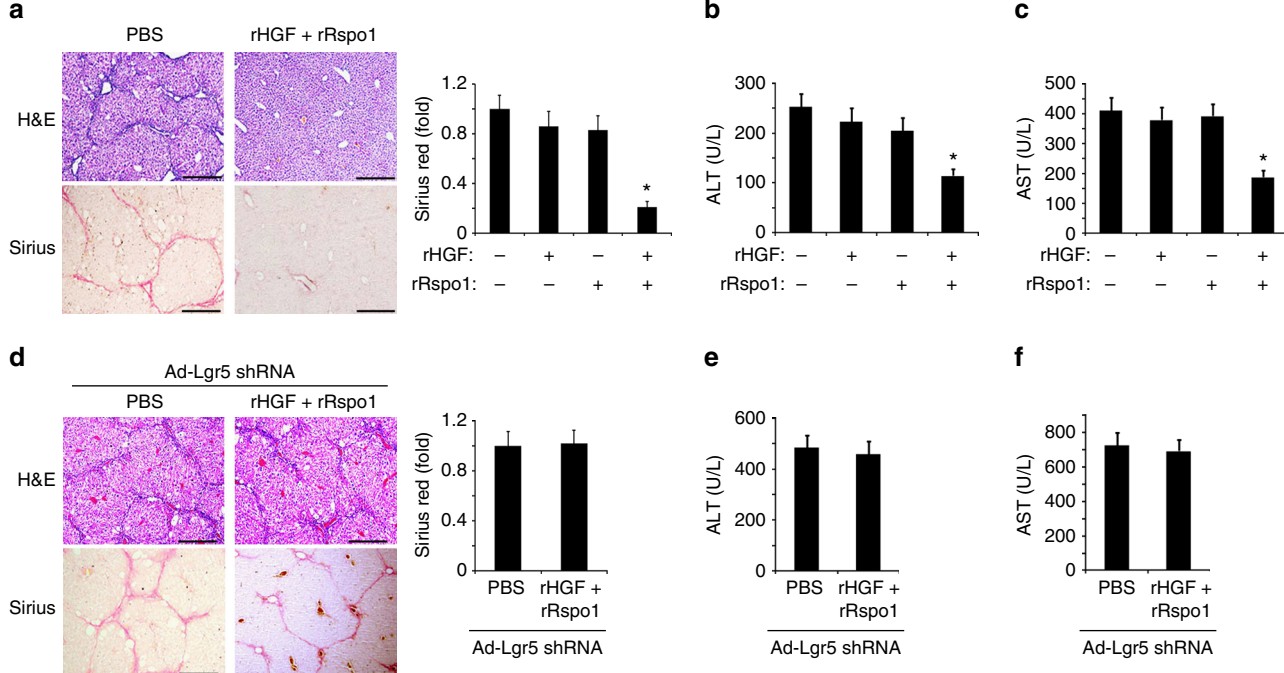

**Fig. 4** rHGF/rRspo1 decreased liver fibrosis through the induction of Lgr5$^+$ cells. **a–c** rHGF/rRspo1 decreased CCL4-induced liver fibrosis and recovered liver functions. CCL4-induced liver fibrosis and rHGF/rRspo1 protein treatments were described in Fig. 3g; the livers were harvested and stained using H&E and Sirius Red for fibrosis analysis (**a**). The serum was harvested for ALT and AST analysis (**b**, **c**). **d–f** rHGF/rRspo1 failed to decrease CCL4-induced liver fibrosis and recover the liver functions when Lgr5 was knocked down. CCL4-induced liver fibrosis and Ad-Lgr5 shRNA infections, and rHGF/rRspo1 protein treatments were described in Supplementary Fig. 15a; the livers were harvested and stained using H&E and Sirius Red for fibrosis analysis (**d**). The serum was harvested for ALT and AST analysis (**e**, **f**). For **a**, **d**, scale bars, 200 μm, the results are shown as mean ± s.d. of five independent sections taken randomly per mice, and a total of 50 tissue specimens in each group ($n = 10$ mice), *$p < 0.05$. For **b**, **c**, **e**, **f**, triplicates for each condition were analyzed. The results are shown as mean ± s.d. of three independent experiments. *$p < 0.05$

tail vein of mice from day 0 to day 4, and the livers were collected and analyzed on day 5 (Fig. 3a). We observed that *Lgr5* mRNA was increased by rHGF and rRspo1 injection alone (Fig. 3b). Furthermore, rHGF in combination with rRspo1 increased the *Lgr5* mRNA levels more effectively when mice liver was injured by CCl4; however, they did not increase the *Lgr5* mRNA levels in oil-treated control mice liver (Fig. 3c). The results of immuno-fluorescence assays with anti-GFP antibody demonstrated that, in control oil-treated *Lgr5-GFP* mice, Lgr5-GFP$^+$ cells were essentially undetectable whether or not the mice were treated with or without recombinant rHGF in combination with rRspo1; however, upon CCL4 treatment, greater numbers of Lgr5-GFP$^+$ cells were observed in the mice treated with rHGF in combination with rRspo1 (Fig. 3d). We then confirmed these results using flow cytometry and sorted the Lgr5-GFP$^+$ liver stem cells from *Lgr5-GFP* mice. We observed that the Lgr5-GFP$^+$ liver stem cells number increased significantly upon treatment with the combination with rRspo1 and rHGF in CCL4-treated mice but not in control oil-treated mice (Fig. 3e, f). In the intestine, rHGF combined with rRspo1 also increased the Lgr5$^+$ intestine stem cell number upon 10 Gy irradiation injury (Supplementary Fig. 14a, b), but failed to increase the Lgr5$^+$ intestine stem cells number in normal mice without irradiation injury (Supplementary Fig. 14c, d).

Next, we tested whether treatment with rHGF combined with rRspo1 increased the number of Lgr5$^+$ liver stem cells during chronic damage. CCL4-induced liver fibrosis model was used here. rHGF and rRspo1 were injected at a dose of 0.1 μg per mice, respectively, into the CCL4-treated *Lgr5-GFP* mice, three times a week for 6 weeks (Fig. 3g). The results of an immunofluorescence

assay with anti-GFP antibody demonstrated significantly more Lgr5-GFP$^+$ cells in mice treated with rHGF combined with rRspo1 (Fig. 3h). We then confirmed these results using flow cytometry and sorted the Lgr5-GFP$^+$ liver stem cells from the *Lgr5-GFP* mice. We observed that the number of Lgr5-GFP$^+$ liver stem cells was significantly increased upon treatment with rHGF combined with rRspo1 compared with the number observed in phosphate-buffered saline (PBS) control-treated mice (Fig. 3i).

To demonstrate whether the Lgr5$^+$ cells numbers, increased by rHGF and rRspo1 combination treatment during liver fibrosis, have liver stem cell properties similar to those of Lgr5$^+$ cells induced by 1XCCL4, single Lgr5$^+$ cells were sorted from *Lgr5-GFP* mice treated with 1XCCl4 alone or 12XCCl4 (liver fibrosis model) plus rHGF and rRspo1. Sorted cells cultured in the stem cell medium conditions rapidly divided and formed organoid structures that were maintained by weekly passaging. Lgr5$^+$ cells from different conditions (1XCCL4 treatment alone or liver fibrosis model plus rHGF and rRspo1 treatment) demonstrated similar abilities to form the organoids, which were in a similar shape, number, and size (Fig. 3j). These results suggested that the administration of recombinant rHGF combined with rRspo1 could increase the number of the Lgr5$^+$ liver stem cells in vivo during chronic damage caused by CCL4.

**rHGF plus rRspo1 attenuates liver fibrosis depending on Lgr5 expression.** We next tested whether rHGF/rRspo1 could recover liver function damaged by CCl4 by increasing the number of Lgr5$^+$ liver stem cells. rHGF or/and rRspo1 were injected at a dose of 0.1 μg per mice into the CCL4-treated C57 mice, three times a week for 6 weeks (Fig. 3g). Using H&E and Sirius Red staining,

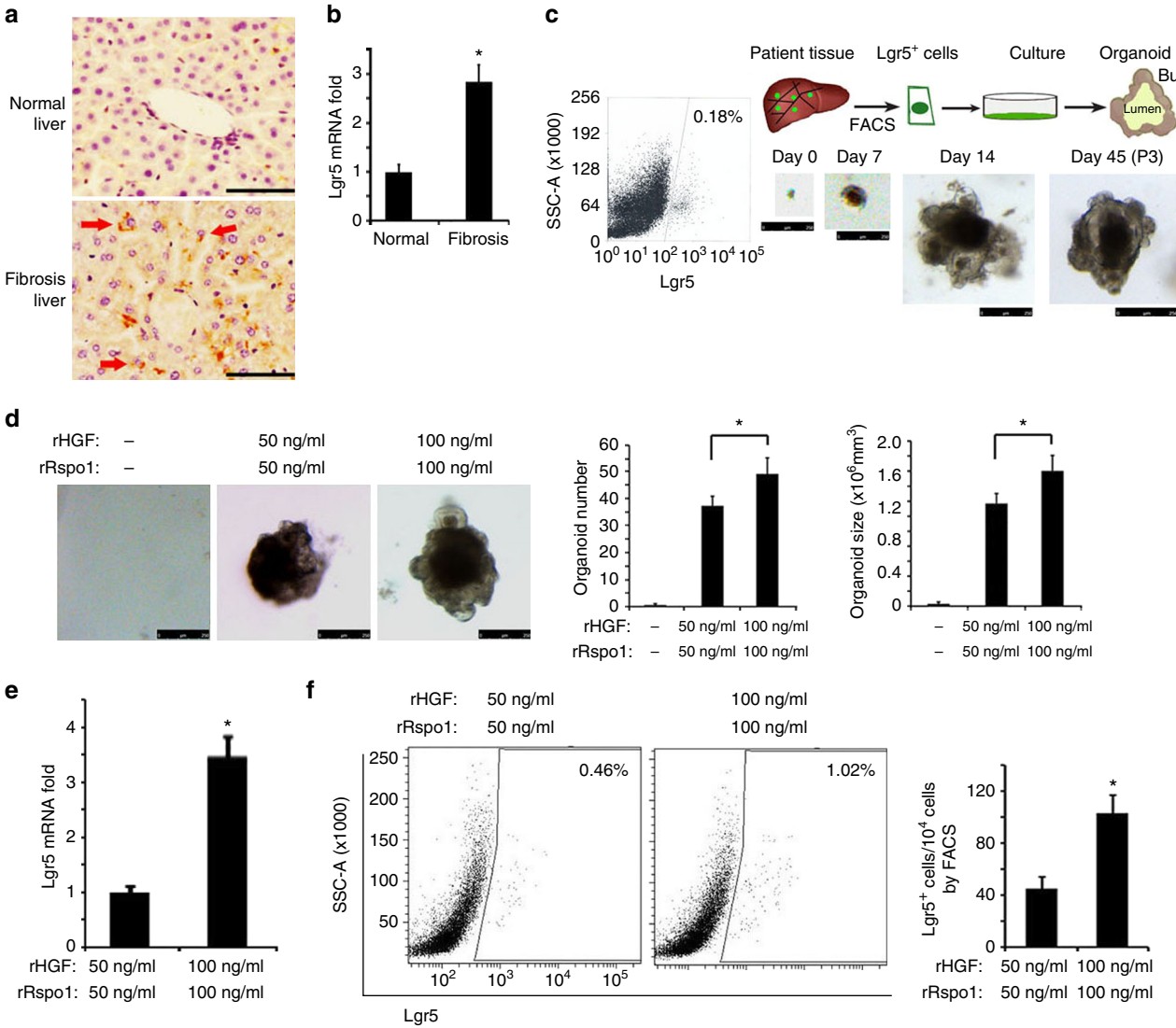

**Fig. 5** Lgr5+ liver stem cells existed in human fibrotic livers. **a** Lgr5 staining in human liver samples using anti-Lgr5 antibody. **b** Lgr5 mRNA level in human liver samples. **c** FACS plot for human Lgr5+ liver cells (left) and a schematic overview of the experimental setup (top right). Single Lgr5+ cells were isolated from six independent donors human liver biopsies diagnosed with liver fibrosis, they were cultured for organoids formation, and continuously cultured for at least three passages in vitro. The representive images were shown (bottom right). **d–f** Single Lgr5+ cells were isolated from six independent donors human liver biopsies diagnosed with liver fibrosis and cultured, and different concentrations of rHGF/rRspo1 were added into the culture medium; the representative organoid pictures, organoids number, and organoids size were analyzed (**d**). The Lgr5 mRNA level in the organoids was measured using qPCR assay (**e**). The Lgr5+ cells numbers were measured using FACS assay (**f**). For **a**, **e**, triplicates for each condition were analyzed. The results are shown as mean ± s.d. of three independent experiments. *$p < 0.05$. For **d**, **f**, the results are shown as mean ± s.d. of three independent experiments. *$p < 0.05$. Scale bars, 200 μm (**a**), 250 μm (**c**, **d**)

we observed that rHGF plus rRspo1 significantly reduced the fibrotic area in the CCL4-induced mice, while rHGF or rRspo1 alone did not demonstrate visible effects (Fig. 4a). The serum ALT and AST levels in the rHGF plus rRspo1-treated mice were also significantly decreased compared with the levels observed in control mice or in mice treated with rHGF or rRspo1 alone (Fig. 4b, c). To verify whether rHGF/rRspo1 recovers the liver functions through Lgr5+ liver stem cells induction, we knocked down Lgr5 expression using Ad-Lgr5 shRNA infection in the CCL4-induced liver fibrosis model. Ad-Lgr5 shRNA or control Ad-Ctrl shRNA was injected into the tail veins of mice tail vein once a week for 6 weeks (Supplementary Fig. 15a). The Lgr5 mRNA levels were decreased by Ad-Lgr5 infection (Supplementary Fig. 15b). We observed that rHGF combined with rRspo1

failed to reduce the liver fibrotic area, as well as serum ALT and AST levels, in CCL4-induced mice infected with Ad-Lgr5 shRNA (Fig. 4d–f), although they significantly reduced the liver fibrotic area and serum ALT, and AST levels in CCL4-induced mice infected with Ad-Ctrl shRNA control (Supplementary Fig. 15c–e).

We next examined whether rHGF/rRspo1 could induce a greater number of Lgr5+ liver stem cells and prevented further progression of established fibrosis. Following the treatment regimen described by Dean Sheppard's lab[21], we treated Lgr5-GFP mice with CCL4 for 3 weeks to establish a fibrotic disease, and then with rHGF/rRspo1 or PBS for the final 3 weeks of CCL4 (Supplementary Fig. 16a). The results of the immunofluorescence assay with anti-GFP antibody demonstrated a significantly greater number of Lgr5-GFP+ cells in the mice treated with rHGF

combined with rRspo1 (Supplementary Fig. 16b). Furthermore, rHGF/rRspo1 significantly reduced liver fibrosis even after a fibrotic disease had been established, as determined by collagen (Sirius red) and a-SMA staining and the hydroxyproline content (Supplementary Fig. 16c–f). The serum levels of ALT and AST (Supplementary Fig. 16g, h) were also significantly reduced upon rHGF/rRspo1 treatment. Furthermore, R-spondin1/HGF treatment showed a more potent effect on reducing liver fibrosis than the agents reported by others (Supplementary Table 1).

**Lgr5$^+$ stem cells are found in human liver fibrosis tissues**. To further investigate whether Lgr5$^+$ liver stem cells exist in the injured human liver, we obtained samples from 42 patients with clinically diagnosed liver fibrosis (Supplementary Table 2), and 5 normal liver samples from automobile accident victims without liver fibrosis to be used as control (Supplementary Table 3). Using an antibody against Lgr5 for immunostaining, we observed that Lgr5 was essentially undetectable in the normal liver, whereas clear Lgr5-positive cells were observed in liver fibrosis samples (Fig. 5a). The enhanced expression of Lgr5 was confirmed by qRT-PCR assay (Fig. 5b). We then tested whether these human Lgr5$^+$ liver cells had stem cell properties. We obtained biopsy samples from five donors with clinically diagnosed liver fibrosis. Single Lgr5$^+$ human liver cells were isolated from these fresh liver fibrosis samples by FACS sorting using a FITC-labeled anti-Lgr5 antibody. Compared with Lgr5-negative cells, the FACS-sorted Lgr5-positive cells expressed a greater number of marker genes of duct-like progenitor cells, including Lgr5, Axin2, Sox9, and CK19, and less or no marker genes of hepatocytes, including Fah, HNF4a, and Alb (Supplementary Fig. 17). Furthermore, sorted cells cultured in stem cells medium[22] rapidly divided and formed organoid structures that were cultured for serial passages (up to four passages; every 2 weeks for each passage) (Fig. 5c).

Next, we investigated whether rHGF/rRspo1 facilitate human Lgr5$^+$ liver stem cells. We added different quantities of rHGF/rRspo1 proteins into the culture medium and observed that single human Lgr5$^+$ liver stem cells could not form an organoid without rHGF/rRspo1 proteins, and that high concentrations of rHGF/rRspo1 proteins helped organoids to grow faster (Fig. 5d), induced more Lgr5 mRNA expression (Fig. 5e), and gave rise to a greater number of human Lgr5$^+$ liver stem cells in the organoids than low concentrations of rHGF/rRspo1 proteins (Fig. 5f). Together, these results indicate that Lgr5$^+$ liver stem cells exist in human liver fibrosis tissues, and that the therapeutic approach for liver fibrosis using rHGF combined with rRspo1 injection to induce more Lgr5$^+$ liver stem cells may also work in humans.

## Discussion

Induction of endogenous adult stem cells by administering soluble molecules provides an advantageous approach for tissue damage repair, which could be a clinically applicable and cost-effective alternative to the transplantation of embryonic or pluripotent stem cell-derived tissues for the treatment of acute organ failures. We previously discovered that Slit2 acts cooperatively with Rspo1 to reduce the intestine's stem cell loss and to alleviate gut injury in response to chemoradiation, leading to a significant prolongation of the overall survival even in animals receiving lethal doses of 5-FU and whole-body/abdominal irradiation[16]. Here, we demonstrated that HGF and R-spondin1 can boost endogenous Lgr5$^+$ liver stem cells to repair livers suffering from chemically induced fibrosis.

Induction of Lgr5$^+$ liver stem cells upon liver damage was first reported by Huch et al.[8] in 2013. Lgr5 is not expressed in the healthy adult liver; however, Lgr5$^+$ cells appear near bile ducts upon acute damage, and these Lgr5$^+$ liver stem cells have been demonstrated to actively contribute to liver regeneration via de novo generation of hepatocytes and ductal cells[8]. In the present study, we demonstrated that Lgr5$^+$ liver stem cells are also induced and retained during chronic fibrotic liver disease. More importantly, Lgr5$^+$ liver stem cell transplantation relieved CCL4-induced liver fibrosis, and when Lgr5 was knocked down, the mice developed more severe fibrosis. Notably, although it has been reported that both the Lgr5 and the Lgr4 genes need to be knocked down simultaneously to remove the stemness from Lgr5$^+$ intestine stem cells[9], we observed that, in Lgr5$^+$ liver stem cells, unlike in Lgr5$^+$ intestine stem cells, there was no Lgr4 proteins expression, and knockdown of Lgr5 alone was enough for Lgr5$^+$ liver stem cells to lose their stemness. This allowed us to deprive Lgr5$^+$ liver stem cells of their functions in vivo by knocking down Lgr5. Furthermore, we observed that the concomitant administration of Rspo1 with the known hepatocyte inducer factor HGF facilitates the induction of Lgr5$^+$ cells in vivo, following damage. In addition, Lgr5$^+$ liver stem cells are also observed in patients diagnosed with liver fibrosis, and rHGF/rRspo1 proteins can increase the number of human Lgr5$^+$ liver stem cells in vitro. Our findings suggested that treatment with HGF/Rspo1 may facilitate liver recovery in a chronic fibrotic liver. This hypothesis is extremely interesting and has huge potential in a clinical setting, considering the fact that liver fibrosis is a worldwide disease that results in the loss of liver function.

It is interesting that HGF/Rspo1 failed to induce Lgr5$^+$ cells in normal livers, which do not have Lgr5$^+$ cells, whereas it induced a greater number of Lgr5$^+$ cells when Lgr5$^+$ cells were induced by liver damage. Lgr5 expression is essential to respond to HGF/Rspo1 treatment to repair liver injury. However, the mechanism by which this takes place needs further study. In addition, the origin of Lgr5$^+$ stem cells remains to be addressed. The potential sources of these damage-induced Lgr5$^+$ stem cells include Lgr5$^-$ progenitor cells from the noninjured liver[23], recruitment from distant sites (e.g., mesenchymal cells)[24], Axin2$^+$ hepatocytes surrounding the central vein[25], Sox9$^+$ hybrid periportal hepatocytes around the portal vein and biliary tree[26], or transdifferentiation of hepatocytes into ductal cells, as that occurs in tumors of the biliary tree (intrahepatic cholangiocarcinoma)[27]. Further studies are needed to explore these hypotheses.

Notably, chronic Wnt/β-catenin activity aberration, including the overexpression of Rspo1 and Lgr5, has recently been reported to promote liver tumorigenesis[28,29]. Moreover, Rspo1 is reported to contribute to hepatic stellate cells (HSC) activation in vitro, and to facilitate the development of liver fibrosis[30]. However, in our studies, Rspo1 injections did not facilitate liver fibrosis in vivo with or without CCL4 damage (Fig. 4a–c, and data not shown). These different results might be due to the differences between in vitro and in vivo conditions. Furthermore, the administration of the combination of rRspo1 and rHGF in our studies is expected to strongly activate Wnt/β-catenin signaling only transiently, which is fundamentally different from the Wnt/β-catenin activation required for tumor development or liver fibrosis development, which takes years. Consequently, it is unlikely that this short pulse of rRspo1/rHGF treatment would increase the risk of developing cancer.

In addition, our unpublished data demonstrated that isolated human Lgr5$^+$ liver stem cells from liver fibrosis samples can be cultured and differentiated into functional mature hepatocytes in vitro. An investigation regarding the potential use of in vitro mature hepatocytes derived from human Lgr5$^+$ liver stem cells in personalized medicine and disease modeling is under way in our lab. However, the mechanism of how Lgr5$^+$ liver stem cells inhibit liver fibrosis formation is still unclear. According to our current data, Lgr5$^+$ liver stem cells induction not only inhibited the development of liver fibrosis but also attenuated the established

fibrosis. New healthy hepatocytes derived from Lgr5[+] liver stem cells probably contributed to liver function recovery. Furthermore, our ongoing studies have demonstrated that Lgr5[+] liver stem cells conditional medium can reverse activated HSCs to quiescent HSCs, which indicates that Lgr5[+] liver stem cells may reduce liver fibrosis through the direct secretion of specific factors. Further studies are needed to explore the detailed mechanisms.

## Methods

**Ethics statement.** The livers without fibrosis were collected from five victims of car accidents with traumatic hepatorrhexis. Debridement of liver trauma was performed for these victims, and the liver tissues were resected for routine pathological biopsy; the samples used in our study were from these biopsy samples. The victims who donated their liver material were not deceased, and all of them provided informed consent retrospectively to use the material for this study. The livers with fibrosis were obtained from 42 patients with clinically diagnosed liver fibrosis, and their liver tissues were collected for routine pathological biopsy; the samples used in our study were from these biopsy samples. All of them provided informed consent retrospectively to use the material for this study. The patients information is provided in Supplementary Tables 2, 3. All study protocols were approved by the Clinical Research Ethics Committee of Southern Medical University and Wenzhou Medical University. Our policy is consistent with the principles embodied in the Declaration of Helsinki. All procedures involving animals, including housing and care, the method by which they were killed, and experimental protocols, were approved by the University Committee on Use and Care of Animals (UCUCA) of Southern Medical University. Our policy is consistent with the U.S. Public Health Service "Policy on Use of Laboratory Animals", available from the Office of Laboratory Animal Welfare, National Institutes of Health.

**Mouse experiments.** Animal experiments were approved by the University Committee on Use and Care of Animals (UCUCA) of Southern Medical University. C57BL6/J (wild type, stock no. 005304), and Lgr5-eGFP-IRES-CreERT2 (Lgr5-GFP, stock no. 008875) mice were purchased from Jackson Laboratory Generation and genotyping of the Lgr5-GFP mice has been previously described[16].

For single CCL4 treatment, 8-week-old male mice received an i.p. injection of CCL4 (2 ml/kg, Sigma-Aldrich) dissolved in olive oil at a ratio of 1:4, or olive oil alone (2 ml/kg). Mice liver was harvested and analyzed at day 5.

For induction of hepatic fibrosis by CCL4, 8-week-old mice were i.p. injected with CCL4 (2 ml/kg, Sigma-Aldrich) dissolved in olive oil at a ratio of 1:4, or olive oil alone (2 ml/kg) twice a week for 6 weeks.

For Sirius Red staining, 5 μM sections were stained with Sirius Red (Sigma), and the results were quantified using Nikon Elements software. Six random fields from each section were analyzed at a final magnification of ×40. Five random sections from each mouse were analyzed, and the statistical data from 10 mice per group were obtained for comparison of the different treatment groups.

**Immunofluorescence.** Liver tissues or liver tumor tissues were fixed with 4% paraformaldehyde, sectioned (5 mm thick), and permeabilized with 0.05% TritonX-100 in PBS, at a pH of 7.4. The samples were blocked with 1% bovine serum albumin (Sigma) and incubated with a primary antibody to GFP (1:200, NB600-303, Novus Biologicals) or Lgr5 (1:100, ab75850, Abcam) at 37 °C for 1 h. After washing extensively, the samples were incubated with the appropriate fluorescent dye-conjugated secondary antibody at 37 °C for 1 h. The sections were counterstained with DAPI or H&E. Slides were then washed and mounted for observation under a scanning confocal microscope (Leica TCS SP2).

**Isolation and quantification of Lgr5[+] liver stem cells.** Livers of Lgr5-GFP mice treated with CCL4 alone or CCL4 plus Rspo1/HGF were collected. Liver cells were isolated using a standard three-step protocol. Liver perfusion was initiated by administering 200 ml of 0.5 M EGTA solution in a basic liver perfusion buffer (30 mM KCl, 1.3 M Nacl, 10 mM NaH$_2$PO$_4$.2H$_2$O, 100 mM glucose, and 100 mM HEPES, pH 7.4) through the portal vein. The liver was then washed with 200 ml of a basic liver perfusion buffer alone. Subsequently, 0.02% collagenase type 4 (Sigma-Aldrich) and 5 mM CaCl$_2$ were added to the basic liver perfusion buffer, and perfusion was continued until digestion was complete. The digested liver was suspended in 50 ml of PBS, and the dissociated cells were passed through a 100-μm nylon mesh and centrifuged at 50×$g$ for 5 min at 4 °C. Single-cell suspension was subjected to flow cytometric sorting, and FL-1(AF488) channel was chosen to gate Lgr5-GFP[+] liver stem cells.

For human samples, fresh liver tissues were cut into small pieces that were resuspended in DMEM consisting of 0.05% trypsin; after a 10-min incubation at 37 °C, the cell suspension was diluted twofold with ice-cold DMEM containing 10% FBS and centrifuged at 300×$g$, and then was resuspended in collagenase buffer consisting of 0.02% Collagenase type IV (Sigma) in DMEM (Gibico). After a 20-min incubation at 37 °C, the cell suspension was passed through a 40-μm cell

strainer and diluted fourfold with ice-cold DMEM containing 10% FBS (Gibico). The cell suspension was centrifuged at 300×$g$ and then, anti-Lgr5 antibody (1:100, ab75850, Abcam) was used to stain the Lgr5[+] cells, followed by an appropriate FITC-conjugated secondary antibody (1:100, E032210-01 EarthOx) for quantification by flow cytometry.

**Recombinant protein.** The recombinant Rspo1 proteins (rRspo1) and HGF proteins (rHGF) are as previously described[16]. Basically, the complimentary DNAs (cDNAs) of human Rspo1 and HGF were amplified for the construction of 6-His fusion proteins, using the forward primer for Rspo1: 5′-TTGCGGCCGCATGCG GCTTGGGCTGTG-3′, and the reverse primer for Rspo1: 5′-GGGAATTCGGCCA GGCCCTGCAGATGTGAGTGGCC-3′; the forward primer for HGF: 5′-TTGCG GCCGCATGTGGGTGACCAAACTCC-3′, and the reverse primer for HGF: 5′-GGGAATTCTGACTGTGGTACCTTATATG-3′. The inserts of Rspo1 and HGF were digested with NotI/EcoRI. They were ligated into the pVL1392 vector (BD Pharmingen).

Recombinant Rspo1 and HGF were expressed in Sf9 insect cells using the baculovirus expression system (BaculoGold; BD Pharmingen) and purified to homogeneity from the serum-free supernatant of Sf9 cells infected with their respective viral stocks (multiplicity of infection, $2 \times 10^8$/ml) by Talon metal affinity chromatography (BD Clontech). Endotoxin levels of these isolated recombinant proteins were <0.1 U/mg of proteins measured by limulus amebocyte lysate (LAL) from Cape Cod. Eluted proteins were dialyzed into PBS buffer.

**qRT-PCR.** qRT-PCR assay was performed as previously described[16,31]. The total RNA of mice livers/tumors or human patient samples was extracted using the Absolutely RNA Miniprep Kit (Stratagene) and reverse transcribed using ThermoScript RT-PCR System (Invitrogen). The resulting cDNA was used for PCR using the SYBR-Green Master PCR Mix (Applied Biosystems) in triplicates. All mouse and human qRT-PCR primer pairs were purchased from SABiosciences. PCR and data collection were performed on the Mx3000 qPCR System (Stratagene). All quantitations were normalized to an endogenous β-actin control. The relative quantitation value for each target gene compared to the calibrator for that target is expressed as $2^{-(Ct-Cc)}$ (Ct and Cc are the mean threshold cycle differences after normalizing to β-actin).

**Culture of Lgr5[+] liver stem cells and organoid culture.** Mice or human Lgr5[+] liver cells were isolated. Single Lgr5-GFP[+] liver cells were mixed with Matrigel (BD Bioscience) and cultured, as described in ref. [8]. The medium composition was as follows: AdDMEM/F12 (Invitrogen) supplemented with 1% B27 and 1% N$_2$ (Invitrogen), N-acetylcysteine (1.25 mM, Sigma-Aldrich), gastrin (10 nM, Sigma), EGF (50 ng/ml, Peprotech), Rspo1 (50 ng/ml, R&D), FGF10 (100 ng/ml, Peprotech), nicotinamide (10 mM, Sigma-Aldrich), and HGF (50 ng/ml, Peprotech). The medium was changed every other day. For human Lgr5[+] liver cells culture, 5 μM A83.01 (Tocris), and 10 μM FSK (Tocris) were added into the medium[22]. For the establishment of the culture, the medium was supplemented with 25 ng/ml Noggin (R&D), 25 ng/ml Wnt (R&D), and 10 μM Y27632 (Sigma-Aldrich) for the first 3 days after isolation. Next, the medium was changed into a medium without Noggin, Wnt, and Y27632. After 10–14 days, organoids were removed from Matrigel, mechanically dissociated into small fragments, and transferred to a fresh matrix. A passage was performed in a 1:4–1:8 split ratio once every 7–10 days for at least 6 months.

**Adenovirus tail vein injection for intrahepatic expression of Lgr5 shRNA.** The custom-made Ad-Lgr5 shRNA (targeting mouse Lgr5) and Ad-nonsense control shRNA were purchased from Vector Biolabs (USA). Lgr5 shRNA target sequence: 5′-TTATTAACAGCAGTCAGGG-3′; control shRNA sequence: 5′-TGAGCAGGCGCATGTGCTG-3′. About $1 \times 10^8$ pfu adenovirus containing, accordingly, construction was injected into the tail vein of the mouse for in vivo Lgr5 knockdown. For in vitro Lgr5 knockdown of Lgr5[+] liver stem cells, $1 \times 10^6$ pfu adenovirus/ml medium was added into the culture medium once a week every week.

**Liver function parameters measurement.** Liver dysfunction was defined according to standard liver biochemistry tests. Serum aspartate aminotransferase (AST) and alanine aminotransferase (ALT) were analyzed using IFCC method on an Automatic Biochemical Analyser (Au2700, Olympus, Japan).

**Hepatocyte functional studies.** To measure glycogen storage and LDL uptake, liver organoids grown in an expansion medium or DM for 14 days were collected, which were stained by periodic acid-Schiff (PAS, Sigma) and DiI-Ac-LDL (Thermo Fisher), respectively, following the manufacturer's instructions. After staining, the organoids were frozen and the frozen sections were obtained for microscope image taking. To determine the secretion of albumin and AAT, liver organoids were differentiated, as previously described. The culture medium was changed every other day, and the culture supernatant was collected on day 14 after differentiation started. Isolated PH (by classical collagenase perfusion) was used as a positive control. The amount of albumin and AAT in the culture supernatant was

determined using a mouse-specific albumin ELISA Quantitation Set (Bethyl Laboratory) and a mouse AAT ELISA Kit (Bethyl Laboratory) according to the manufacturer's instructions.

**Lgr5+ liver stem cell transplantation**. *Lgr5-GFP* mice liver stem cells were isolated by FACS sorting and then incubated on ice. The recipient mice (hepatic fibrosis model, 8-week-old mice were i.p. injected with CCL4 (2 ml/kg, Sigma-Aldrich) dissolved in olive oil at a ratio of 1:4, or olive oil alone (2 ml/kg) twice a week for 6 weeks) were anesthetized using 1% pentobarbital sodium (1 ml/kg) i.p., and the hair on the left side of the abdomen was removed. A superficial 1.0-cm skin incision was made and the spleen was revealed. Approximately $1 \times 10^6$ Lgr5+ liver stem cells or PH was injected into the spleen subcutaneously on day 0. One-hundred microliters of penicillin and streptomycin (Gibico, Cat:15140122) were injected in the abdomen before suturing the wound. On day 40, the mice were killed to collect blood and liver samples.

**Western blotting of organoids**. Liver organoids were grown in an expansion medium for 10 days, and $1 \times 10^8$ pfu of Ad-Lgr5 shRNA or Ad-Ctrl shRNA were added into the medium. After 3 days, the organoids were collected, incubated with 200 μl of lysis buffer (Tris-HCL 50 mM, pH 7.4, Nacl 150 mM, sodium deoxycholate 0.25%, NP-40 1%, EDTA 1 mM, PMSF 1 mM, aprotinin 1 μg/ml, leupeptin 1 μg/ml, and pepstain 1 μg/ml) on ice for 30 min, and centrifuged at 12,000 rpm for 5 min. The supernatant was boiled with 5× loading buffer for 5 min followed by SDS-PAGE and western blotting using the following antibodies: anti-Lgr5 (1:2000, ab75850, Abcam), and anti-β-actin (1:2000, A5441, Sigma).

**Patients and immunohistochemical staining**. Human liver biopsies (0.5–1 cm³) were obtained from a donor at Taizhou Hospital and Nanfang Hospital. Informed consent was obtained from each patient, and the study protocol was approved by the Clinical Research Ethics Committee of Wenzhou Medical University and Southern Medical University. A diagnosis of liver fibrosis was confirmed on histologic examination.

The immunohistochemical staining was performed as before[16,31]. About 5-μm-thick tissues sections were obtained and incubated with primary anti-Lgr5 antibodies (1:100, ab75850, Abcam), and then followed by incubation with a horseradish peroxidase anti-rabbit IgG antibody. The color was then developed by incubation with DAB Substrate Kit (Pierce). After washing, the tissue sections were finally counterstained with hematoxylin.

**Knockdown of Lgr5 by CRISPR/Cas9 system**. The adenovirus CRISPR-Cas9 system for Lgr5 knockout was performed by Obio Technology (Shanghai) Corp., Ltd. Briefly, the cDNA of Cas9 and the gRNA, the target of which was designed to the first exon of the Lgr5 gene, was constructed into the adenovirus vector. Control nonsense gRNA sequence: 5′-ACACTGTCACTGTGAGCTGGATGG-3′. The construct was verified by DNA sequence analysis. For isolated Lgr5+ liver stem cells culture in vitro, $1 \times 10^6$ pfu adenovirus containing, accordingly, construction was added into 1 ml of culture medium once a week. For the mice model in vivo, $1 \times 10^8$ pfu adenovirus containing, accordingly, construction was injected into the tail vein of a mouse once a week for 6 weeks.

**Statistics**. The experimental data were statistically analyzed by Student's *t* test or Mann–Whitney *U* test. All data are presented as mean ± s.d. $p < 0.05$ was considered as statistically significant. In all cases, data from at least three independent experiments were used. All calculations were performed using SPSS software package. No randomization was used. No blinding was done. Power calculations were not done to predetermine sample sizes.

**Data availability**. All relevant data are available from the authors.

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

## Acknowledgements

This work was supported by Thousand Youth Talents Plan Project, The National Science Foundation of China (81472313, 81573015, 81600496, and 81672821), Guangdong Provincial Natural Science Foundation for Distinguished Young Scientists (2015A030306048), Guangzhou Science and Technology Collaborative Innovation Major Projects (201704020071), The National Basic Research Program of China (973 program, 2015CB554002), Project of the National Natural Science Foundation of China supported by NSFC-Guangdong Joint Fund (U1201226), and the National Key R&D Program of China (2017YFC1309002).

## Author contributions

W.-J.Z. conceived the project. Y.L. and W.-J.Z. designed the experiments. Y.L. and Z.-P.F. performed all the liver stem cells isolation and culture experiments and analyzed the data. Y.L., Z.-P.F., H.-J.L., L.-J.W., Z.C., N.T., T.L., H.-X.H. and G.C. performed analyses of the

human patient samples and the mouse models. W.-J.Z., Y.-Q.D. and L.L. supervised the project and wrote the manuscript. All authors read and approved the final manuscript.

## Additional information

**Competing interests:** The authors declare no competing financial interests.

