## [Peer Review file · Nature Communications]

Editorial Note: This manuscript has been previously reviewed at another journal that is not operating a transparent peer review scheme. This document only contains reviewer comments and rebuttal letters for versions considered at Nature Communications. Mentions of prior referee reports have been redacted.

REVIEWERS' COMMENTS:

Reviewer #1 (Remarks to the Author):

In this particular manuscript the authors demonstrate that HGF and R-Spondin1 can recruit endogenous liver stem cells to repair livers suffering from chemically induced fibrosis. The manuscript builds on previous studies that have demonstrated a beneficial role for HGF in liver injury and for R-spondin-1 in recovery from tissue insults such as graft versus host-disease, experimental colitis, and intestinal injury, through the activation of endogenous stem cells. The main findings of the manuscript are that:

- 1: Lgr5 expression is required for the stem cell response in the liver fibrosis model
- 2: R-spondin-I and HGF boost the Lgr5+ stem cell response.
- 2: Lgr5+ cells that can give rise to organoid cultures are present in human fibrotic livers

The experiments are generally well-designed and properly controlled and justify the conclusions drawn by the authors. Notable exceptions are immunofluorescent and histochemical images that frequently lack nuclear stains and/or scale bars. Some immunohistochemical images have been quantified although they appear to have been taken with different microscope settings (e.g. Supplemental figure 10SE). The manuscript is generally well-written, although additional language editing is necessary and the materials and methods section is far from complete. The manuscript would also improve if the authors could give more insight into the mechanism through which Lgr5+ cells prevent liver fibrosis. If the authors are able to address my major comments, am in favour of publication of this manuscript in Nature Communications.

Major comments

- Some immunohistochemical images have been quantified although they appear to have been taken with different microscope settings (e.g. Supplemental figure 10SE top right and lower right appear to have different contrast/light scattering). The authors should be able to explain this discrepancy between these images and they should be able to take away any doubts that these differences in any way influence the quantification.
- The authors have used liver material without fibrosis from victims of car accidents as control tissues. They indicate that these patients have given informed consent to use their liver tissue for research. To me, victims of car accidents do not appear to be in the best position to make a well-thought voluntary decision on giving informed consent. Can the authors give more insight in their informed consent procedure to assure that all material was obtained according to ethical standards as defined by for example Nature Publishing Group.
- For essential materials and methods, such as the culture and differentiation conditions of mouse and human liver stem cells, the authors refer to papers from other groups. I am in favour of at least providing a short description of the culture conditions that were used. The authors do not provide any information on how Albumin, AAT, glycogen, and LDL levels were measured. I also could not find any information on how the intrasplenic injections were performed, nor on how western blotting and immunohistochemistry on organoids was performed. How were the recombinant proteins produced and purified? The authors should carefully check if the materials and methods of all other experiments are covered in the M&M section. For some reason or another, the QPCR experiments are discussed twice in the M&M section, so space limitations is not

a valid argument not to include other methods.

- Can the authors provide a mechanistic model (e.g. in their discussion section) for how Lgr5+ cells prevent liver fibrosis, because these are unlikely to be the cell type of origin for the hepatic myofibroblasts, which instead are thought to originate from hepatic stellate cells and portal fibroblasts? This would be highly valuable for future studies.
- The authors claim that the Lgr5+ cells are similar to those described by Huch et al. Nevertheless, the morphology of the organoids derived from these Lgr5+ cells (figs 2B, 3J, S4A, S8, S10C) is remarkably different from those described by Huch et al in 2013 and 2015. Can the authors explain this difference?

Minor comments

- The manuscript would benefit from language editing, e.g.
 - o Abstract: replace "...into failing organs for treating acute organ failures..." with "... for treating acute organ failures ..."
 - o Abstract: replace "...protect the intestine and prolong overall survival in mice through the induction of LGR5+ intestine cells"
 - o Abstract: replace "...liver fibrosis caused by CCl4..." with "...liver fibrosis caused by CCl4..." (this typo occurs frequently throughout the entire manuscript, e.g. last paragraph at page 5 introduction and in subheading on page 8)
 - o Final sentence of the introduction: I would use present tense instead of past tense in this sentence.
 - o Results section: remove or modify the floating sentence: "The CCL4-induced mouse model of liver fibrosis was built"
 - o Results section page 9: homolog should be homologue
 - o Results section page 9: rephrase: "That indicated we could use Ad-Lgr5shRNA to make the Lgr5+ liver stem cells to lose their stem cells abilities for further study."
 - o etc.
- Abstract: "...induction of Lgr5+ intestine stem cells." Is this upon damage?
- Abstract: "HGF plus RSPO1 proteins injections induced significantly more ..." Do the authors here mean that there is a synergistic effect between HGF and RSPO1 that is higher than the individual effects of HGF and RSPO1 alone?
- Introduction page 4 first sentence: "some other animals" is unspecific
- Results section page 6: "We used Lgr5-enhanced green fluorescent protein (eGFP)-internal ribosome entry site (IRES)-CreERT2 (Lgr5-GFP) mice (16) to detect the expression of Lgr5-GFP in liver." Here, the authors refer to their own paper. To my knowledge this mouse model was built and first described by the Clevers lab in 2007 (ref 6). Please refer to this study.
- Results section page 9: replace "...destruct..." with "...inhibit..."
- Fig 3e: percentages are missing in the 3rd FACS plot
- Supplemental figure S2: Lgr5 mRNA levels stabilize after the first 5 days of CCL4 administration. What is the reason for this stabilization? Is there a static population of Lgr5 positive cells from day 5 onwards or is there a continuous recruitment/supply of Lgr5 positive cells (for example through stem cell divisions) that is counterbalanced by a continuous loss of Lgr5+ cells (for example through differentiation/cell death)? This is important because this may help understand the mechanism through which the Lgr5+ cells counter liver fibrosis.
- Supplemental figure S4 panels g and h: these figures look like tissue sections to me, not like stainings on organoids.
- Supplemental figure S5: I find these images unconvincing: Lgr5 and SOX9 have complete overlap, even though Lgr5 GFP should have cytoplasmic expression (as is the case in supplemental figs 3A and 11B) and SOX9 should be nuclear. A nuclear stain is also missing in this picture, as are the scale bars even though these are mentioned in the legend.
- Supplemental figure S13a and b: can the authors explain why there are no GFP+ cells in the intestine of the control condition (not treated with HGF and RSPO1). Are all Lgr5+ intestinal stem cells lost after radiation?

- Discussion section first paragraph: the discussion of ESCs and iPSCs seems irrelevant to this particular study

Ewart Kuijk

Reviewer #2 (Remarks to the Author):

Overall, this is an improved version of this manuscript where my previous concerns are addressed. It is an interesting manuscript that has translational potential as the results obtained might indicate that Lgr5+ liver stem cells might have the potential to alleviate liver fibrosis in chronic liver damage. I have only 2 comments:

1) in Fig 1b the authors mention that the GFP staining identified the transplanted Lgr5+ population. At what time point after transplantation is this done? Similarly, what is not addressed is what cell types do these Lgr5+ cells contribute to after transplant. Do they only generate hepatocytes? Ductal cells? Other cells? In the absence of lineage tracing maybe the authors can speculate on their discussion?

2) Another question that arises from this manuscript, is the molecular mechanism by which the injection of these Lgr5+ cells results in a reduction of liver fibrosis. Is the fibrosis cleared or simply not happening? While that could be a follow-up story out of the scope of this manuscript, it would be interesting for the authors to discuss their thoughts on that on the discussion section.

Point-by-point response (in blue) to Reviewer's comments:

REVIEWERS' COMMENTS:

Reviewer #1 (Remarks to the Author):

In this particular manuscript the authors demonstrate that HGF and R-Spondin1 can recruit endogenous liver stem cells to repair livers suffering from chemically induced fibrosis. The manuscript builds on previous studies that have demonstrated a beneficial role for HGF in liver injury and for R-spondin-1 in recovery from tissue insults such as graft versus host-disease, experimental colitis, and intestinal injury, through the activation of endogenous stem cells. The main findings of the manuscript are that:

- 1: Lgr5 expression is required for the stem cell response in the liver fibrosis model
- 2: R-spondin-1 and HGF boost the Lgr5+ stem cell response.
- 2: Lgr5+ cells that can give rise to organoid cultures are present in human fibrotic livers

The experiments are generally well-designed and properly controlled and justify the conclusions drawn by the authors. Notable exceptions are immunofluorescent and histochemical images that frequently lack nuclear stains and/or scale bars. Some immunohistochemical images have been quantified although they appear to have been taken with different microscope settings (e.g. Supplemental figure 10SE). The manuscript is generally well-written, although additional language editing is necessary and the materials and methods section is far from complete. The manuscript would also improve if the authors could give more insight into the mechanism through which Lgr5+ cells prevent liver fibrosis. If the authors are able to address my major comments, am in favour of publication of this manuscript in Nature Communications.

We thank the reviewer for the positive comments on our study. We are greatly encouraged! We also highly appreciate the insightful critics aimed at improving

and solidifying our study.

In revised version, the immunohistochemical images have been improved, and we also asked Nature Publishing Group Language Editing service to review and edit our manuscript. The materials and methods section is also well completed now. We discussed the mechanism through which Lgr5⁺ cells prevent liver fibrosis in discussion section:

“According to our current data, Lgr5⁺ liver stem cells induction not only inhibited the development of liver fibrosis but also attenuated established fibrosis. New healthy hepatocytes derived from Lgr5⁺ liver stem cells probably contributed to liver function recovery. Furthermore, our ongoing studies have demonstrated that Lgr5⁺ liver stem cells conditional medium can reverse activated hepatic stellate cells (HSCs) to quiescent HSCs, which indicates that Lgr5⁺ liver stem cells may reduce liver fibrosis through the direct secretion of specific factors. Further studies are needed to explore the detailed mechanisms.”

Major comments

- Some immunohistochemical images have been quantified although they appear to have been taken with different microscope settings (e.g. Supplemental figure 10SE top right and lower right appear to have different contrast/light scattering). The authors should be able to explain this discrepancy between these images and they should be able to take away any doubts that these differences in any way influence the quantification.

We thank the reviewer for pointing this out for us. The immunohistochemical images in supplementary figure 10SE top right and lower right (in revised version, they are isupplementary figure 11E) had been improved using new images. We used the same microscope settings to ensure images with high quantification.

- The authors have used liver material without fibrosis from victims of car accidents as control tissues. They indicate that these patients have given informed consent to use their liver tissue for research. To me, victims of car accidents do not appear to be in the best position to make a well-thought voluntary decision on giving informed consent. Can the authors give more insight in their informed consent procedure to assure that all material was obtained according to ethical standards as defined by for example Nature Publishing Group.

We explain the sequence of events in revised version:

“The livers without fibrosis were harvested from 5 victims of car accidents with traumatic hepatorrhesis. Debridement of liver trauma was performed for these victims, and the liver tissues were resected for routine pathological biopsy; the samples used in our study were from these biopsy samples. The victims who donated their liver material were not deceased, and all of them provided informed consent retrospectively to use the material for this study. All study protocols were approved by the Clinical Research Ethics Committee of Southern Medical University and Wenzhou Medical University. Our policy is consistent with the principles embodied in the Declaration of Helsinki. “

- For essential materials and methods, such as the culture and differentiation conditions of mouse and human liver stem cells, the authors refer to papers from other groups. I am in favour of at least providing a short description of the culture conditions that were used.

We thank the reviewer for pointing this out for us. We added the detailed method accordingly into the Materials & Methods section.

“Culture of Lgr5⁺ liver stem cells and organoid culture

Mice or human Lgr5⁺ liver cells were isolated. Single Lgr5-GFP⁺ liver cells were mixed with Matrigel (BD Bioscience) and cultured as described in ref. 8. Medium composition was as follows: AdDMEM/F12 (Invitrogen) supplemented

with 1% B27 and 1% N2 (Invitrogen), N-acetylcysteine (1.25 mM, Sigma-Aldrich), gastrin (10 nM, Sigma), EGF (50 ngml⁻¹, Peprotech), Rspo1 (50 ngml⁻¹, R&D), FGF10 (100 ngml⁻¹, Peprotech), nicotinamide (10mM, Sigma-Aldrich) and HGF (50 ng/ml, Peprotech). Medium was changed every other day. For human Lgr5+ liver cells culture, 5 μM A83.01 (Tocris), and 10 μM FSK (Tocris) were added into the medium²². For the establishment of the culture, the medium was supplemented with 25 ng/ml Noggin (R&D), 25 ng/ml Wnt (R&D), and 10 μM (Y27632, SigmaAldrich). for the first 3 days after isolation. Next,, the medium was changed into a medium without Noggin, Wnt, Y27632. After 10 - 14 days, organoids were removed from the Matrigel, mechanically dissociated into small fragments, and transferred to fresh matrix. Passage was performed in a 1:4 - 1:8 split ratio once every 7 - 10 days for at least 6 months.”

The authors do not provide any information on how Albumin, AAT, glycogen, and LDL levels were measured. I also could not find any information on how the intrasplenic injections were performed, nor on how western blotting and immunohistochemistry on organoids was performed. How were the recombinant proteins produced and purified? The authors should carefully check if the materials and methods of all other experiments are covered in the M&M section. For some reason or another, the QPCR experiments are discussed twice in the M&M section, so space limitations is not a valid argument not to include other methods.

We thank the reviewer for pointing this out for us. The materials and methods section is well completed in revised version. We did not perform immunohistochemistry on organoids in this study, the other missing information are as following:

Hepatocyte functional studies

To measure glycogen storage and LDL uptake, liver organoids grown in expansion medium or differentiation medium for 14 days were collected which were stained by periodic acid-Schiff (PAS, Sigma) and Dil-Ac-LDL (Thermo Fisher), respectively, following the manufacturer’s instructions. After staining, the organoids were frozen and the frozen sections were gotten for microscope image taking. To determine the secretion of albumin and AAT, liver organoids were differentiated as previously described. Culture medium was changed every other day and culture supernatant was collected on day 14 after differentiation started. Isolated primary hepatocytes (by classical collagenase perfusion) were used as positive controls. The amount of albumin and AAT in the culture supernatant was determined using a mouse-specific albumin ELISA Quantitation set (Bethyl Laboratory) and a mouse AAT ELISA kit (Bethyl Laboratory) according to the manufacturer’s instructions.

Lgr5⁺ liver stem cell transplantation

Lgr5–*GFP* mice liver stem cells were isolated by FACS sorting and then incubated on ice. The recipient mice (hepatic fibrosis model, 8-week old mice were i.p. injected with CCL4 (2 mlkg⁻¹, Sigma-Aldrich) dissolved in olive oil in a ratio of 1:4, or olive oil alone (2 mlkg⁻¹) twice a week for 6 weeks) were anesthetized using 1% pentobarbital sodium (1mlkg⁻¹) i.p., and the hair on the left side of the abdomen was removed. A superficial 1.0 cm skin incision was made and the spleen was revealed. Approximately 1x10⁶ Lgr5⁺ liver stem cells or primary hepatocytes were injected into the spleen subcutaneously on day 0. One hundred microliters of penicillin and streptomycin (Gibico, Cat: 15140122) were injected in the abdomen before suturing the wound. On day 40, mice were sacrificed to collect blood and liver samples.

Western blotting of organoids

Liver organoids were grown in expansion medium for 10 days, and 1x10⁸ pfu of Ad-Lgr5 shRNA or Ad-Ctrl shRNA were added into the medium, after 3 days, the organoids were collected, incubated with 200 µl lysis buffer (Tris-HCL 50mM, PH 7.4, NaCl 150 mM, sodium deoxycholate 0.25%, NP-40 1%, EDTA 1 mM, PMSF 1 mM, Aprotinin 1µgml⁻¹, leupeptin 1µgml⁻¹, pepstain 1µgml⁻¹) on ice for 30 min, and centrifuged at 12000 rpm for 5 min. The supernatant was boiled with 5x loading buffer for 5 min followed by SDS-PAGE and western blotting using the following antibodies: anti-Lgr5 (1:2000, ab75850, Abcam), anti-β-actin (1:2000, A5441, Sigma).

Recombinant protein

The Recombinant Rspo1 proteins (rRspo1) and HGF proteins (rHGF) are as previously described [16]. Basically, the cDNAs of human Rspo1 and HGF were amplified for construction of 6-His fusion proteins, using the forward primer for Rspo1: 5'-TTGCGGCCGCATGCGGCTTGGGCTGTG-3', and the reverse primer for Rspo1: 5'-GGGAATTCGGCAGGCCCTGCAGATGTGAGTGGCC-3'; the forward primer for HGF: 5'-TTGCGGCCGCATGTGGGTGACCAAACCTCC-3', and the reverse primer for HGF: 5'-GGGAATTCTGACTGTGGTACCTTATATG-3'. The inserts of Rspo1 and HGF were digested with NotI/EcoRI. They were ligated into the pVL1392 vector (BD Pharmingen).

Recombinant Rspo1 and HGF were expressed in Sf9 insect cells using the bacu-lovirus expression system (BaculoGold; BD Pharmingen) and purified to homo-geneity from the serum-free supernatant of Sf9 cells infected with their respective Viralstocks (multiplicity of infection, 2x10⁸ ml⁻¹) by Talon metal affinity chro-Matography (BD Clontech). Endotoxin levels of these isolated recombinant proteins were < 0.1 U mg⁻¹ of proteins measured by limulusa moebocytelysate (LAL) from Cape Cod. Eluted proteins were dialyzed into

PBS buffer.

- Can the authors provide a mechanistic model (e.g. in their discussion section) for how Lgr5⁺ cells prevent liver fibrosis, because these are unlikely to be the cell type of origin for the hepatic myofibroblasts, which instead are thought to originate from hepatic stellate cells and portal fibroblasts? This would be highly valuable for future studies.

We thank the reviewer for the valuable suggestions. We discussed the mechanism through which Lgr5⁺ cells prevented liver fibrosis in discussion section:

“According to our current data, Lgr5⁺ liver stem cells induction not only inhibited the development of liver fibrosis but also attenuated established fibrosis. New healthy hepatocytes derived from Lgr5⁺ liver stem cells probably contributed to liver function recovery. Furthermore, our ongoing studies have demonstrated that Lgr5⁺ liver stem cells conditional medium can reverse activated hepatic stellate cells (HSCs) to quiescent HSCs, which indicates that Lgr5⁺ liver stem cells may reduce liver fibrosis through the direct secretion of specific factors. Further studies are needed to explore the detailed mechanisms”

- The authors claim that the Lgr5⁺ cells are similar to those described by Huch et al. Nevertheless, the morphology of the organoids derived from these Lgr5⁺ cells (figs 2B, 3J, S4A, S8, S10C) is remarkably different from those described by Huch et al in 2013 and 2015. Can the authors explain this difference?

We thank the reviewer for pointing this out for us.

Functionally, our Lgr5⁺ cells sorted from liver fibrosis model mice cultured in the stem cells medium conditions rapidly divided and formed organoid structures that were maintained by weekly passaging. When the Lgr5⁺ cells were cultured in differentiation medium (DM), they expressed mature hepatic genes, and abundant amounts of albumin and AAT were secreted into the medium. The differentiated cells were competent for accumulated glycogen and low-density lipoprotein (LDL) uptake. All these functions are similar to those described by Huch et al. induced upon 1X CCl₄. However, we agree that the morphology of the organoids is different from those described by Huch et al., that might due to the different culture environment and different microscope settings when taking photo.

To avoid confusion, we modified our statement, we replaced “We next asked whether these Lgr5⁺ cells induced upon chronic damage are similar cells to the

ones described in Huch et al. induced upon 1X CCl4.” to “Next, we investigated whether Lgr5⁺ cells induced upon chronic damage are liver stem cells.” ; replace “These results suggested that these Lgr5⁺ cells induced upon chronic damage are similar cells to the ones described in Huch et al. induced upon 1X CCl4.” to “These results suggest that these Lgr5⁺ cells that are induced upon chronic damage are liver stem cells.”

Minor comments

- The manuscript would benefit from language editing, e.g.

We thank the reviewer for the valuable suggestion. We have asked Nature Publishing Group Language Editing service to review and edit our manuscript. Our current version would be clear and well written.

- o Abstract: replace “into failing organs for treating acute organ failures” with “for treating acute organ failures”
- o Abstract: replace “protect the intestine and prolong overall survival in mice through the induction of LGR5⁺ intestine cells”
- o Abstract: replace “liver fibrosis coursed by CCl4” with “liver fibrosis caused by CCl4” (this typo occurs frequently throughout the entire manuscript, e.g. last paragraph at page 5 introduction and in subheading on page 8)

We thank the reviewer for pointing these out for us. We used the edited abstract by the editor in our revised manuscript.

- o Final sentence of the introduction: I would use present tense instead of past tense in this sentence.

We thank the reviewer for pointing these out for us. We used present tense in this sentence in our revised manuscript according to the reviewer’s suggestion.

“These findings indicate that Lgr5⁺ liver stem cells are crucial for recovery from liver dysfunction and that the combination of of HGF and Rspo1 which induces Lgr5⁺ stem cells, might be be able to be used for liver fibrosis therapy.”

- o Results section: remove or modify the floating sentence: “The CCl4-induced mouse model of liver fibrosis was built”

We thank the reviewer for pointing these out for us. We removed this sentence in our revised manuscript according to the reviewer’s suggestion.

- o Results section page 9: homolog should be homologue

We thank the reviewer for pointing these out for us. “Homolog” have been corrected to “homologue” in the revised manuscript.

o Results section page 9: rephrase: “That indicated we could use Ad-Lgr5shRNA to make the Lgr5⁺ liver stem cells to lose their stem cell abilities for further study.”

We thank the reviewer for pointing these out for us. We edited this sentence to: “the results indicated that Ad-Lgr5shRNA can be used to mimic Lgr5⁺ liver stem cell lose-of-function for further study.”

o etc.

We have asked Nature Publishing Group Language Editing service to review and edit our manuscript. Our current version would be clear and well written.

• Abstract: “...induction of Lgr5⁺ intestine stem cells.” Is this upon damage?

Yes, “...induction of Lgr5⁺ intestine stem cells upon chemoradiotherapy.”

• Abstract: “HGF plus RSP01 proteins injections induced significantly more ...” Do the authors here mean that there is a synergistic effect between HGF and RSP01 that is higher than the individual effects of HGF and RSP01 alone?

No, here we mean HGF plus RSP01 treatment induced more Lgr5⁺ liver stem cells than negative control.

• Introduction page 4 first sentence: “some other animals” is unspecific

We thank the reviewer for pointing these out for us, we deleted “some other animals” and now the sentence is: “The liver is a vital organ of the digestive system in vertebrates.”

• Results section page 6: “We used Lgr5-enhanced green fluorescent protein (eGFP)-internal ribosome entry site (IRES)-CreERT2 (Lgr5-GFP) mice (16) to detect the expression of Lgr5-GFP in liver.” Here, the authors refer to their own paper. To my knowledge this mouse model was built and first described by the Clevers lab in 2007 (ref 6). Please refer to this study.

We thank the reviewer for pointing these out for us, we corrected this mistake and refer to paper of the Clevers lab in 2007.

- Results section page 9: replace "...destruct..." with "...inhibit..."

We thank the reviewer for pointing these out for us, we have replaced replace "...destruct..." with "...inhibit..." accordingly.

- Fig 3e: percentages are missing in the 3rd FACS plot

We thank the reviewer for pointing these out for us, the percentages have been added accordingly.

- Supplemental figure S2: Lgr5 mRNA levels stabilize after the first 5 days of CCL4 administration. What is the reason for this stabilization? Is there a static population of Lgr5 positive cells from day 5 onwards or is there a continuous recruitment/supply of Lgr5 positive cells (for example through stem cell divisions) that is counterbalanced by a continuous loss of Lgr5+ cells (for example through differentiation/cell death)? This is important because this may help understand the mechanism through which the Lgr5+ cells counter liver fibrosis.

We thank the reviewer for this insightful comment. We found that the Lgr5 expression induced by single dose of CCL4 was reduced after liver recovery (Supplementary Fig. 1). So it seemed Lgr5 would be continuous existence until liver recovery, that means, continuous injury induced continuous Lgr5+ cells existing. We have discussed the origin of Lgr5+ cells, and also discussed the mechanism through which the Lgr5+ cells counter liver fibrosis.

“In addition, the origin of Lgr5⁺ stem cells remains to be addressed. Potential sources of these damage-induced Lgr5⁺ stem cells include Lgr5⁻ progenitor cells from the noninjured liver (23), recruitment from distant sites (e.g., mesenchymal cells) (24), Axin2⁺ hepatocytes surrounding the central vein (25), Sox9⁺ hybrid periportal hepatocytes around the portal vein and biliary tree (26), or transdifferentiation of hepatocytes into ductal cells, as occurs in tumors of the biliary tree (intrahepatic cholangiocarcinoma) (27). Further studies are needed to explore these hypotheses.”

“According to our current data, Lgr5⁺ liver stem cells induction not only inhibited the development of liver fibrosis but also attenuated established fibrosis. New healthy hepatocytes derived from Lgr5⁺ liver stem cells probably contributed to liver function recovery. Furthermore, our ongoing studies have demonstrated that Lgr5⁺ liver stem cells conditional medium can reverse activated hepatic stellate cells (HSCs) to quiescent HSCs, which indicates that

Lgr5⁺ liver stem cells may reduce liver fibrosis through the direct secretion of specific factors. Further studies are needed to explore the detailed mechanisms.”

- Supplemental figure S4 panels g and h: these figures look like tissue sections to me, not like stainings on organoids.

These images were taken from organoid frozen sections but not whole-mount organoids, that is why they look like tissue sections. The staining method were described in Material and Method section.

“To measure glycogen storage and LDL uptake, liver organoids grown in expansion medium or differentiation medium for 14 days were collected which were stained by periodic acid-Schiff (PAS, Sigma) and Dil-Ac-LDL (Thermo Fisher), respectively, following the manufacturer’s instructions. After staining, the organoids were frozen and the frozen sections were gotten for microscope image taking. “

- Supplemental figure S5: I find these images unconvincing: Lgr5 and SOX9 have complete overlap, even though Lgr5 GFP should have cytoplasmic expression (as is the case in supplemental figs 3A and 11B) and SOX9 should be nuclear. A nuclear stain is also missing in this picture, as are the scale bars even though these are mentioned in the legend.

We thank the reviewer for pointing these out for us, the restained and improved images were used now, in current figure, Lgr5 GFP mostly cytoplasmic expression and SOX9 mostly nuclear expression, DAPI was used for nuclear staining.

- Supplemental figure S13a and b: can the authors explain why there are no GFP⁺ cells in the intestine of the control condition (not treated with HGF and RSP01). Are all Lgr5⁺ intestinal stem cells lost after radiation?

Most Lgr5⁺ intestinal stem cells lost after radiation, this result is consistent with our previous report (Zhou et al, Nature, 2013)

- Discussion section first paragraph: the discussion of ESCs and iPSCs seems irrelevant to this particular study

We thank the reviewer for the valuable suggestions. We deleted the discussion of ESCs and iPSCs accordingly.

Ewart Kuijk

Reviewer #2 (Remarks to the Author):

Overall, this is an improved version of this manuscript where my previous concerns are addressed. It is an interesting manuscript that has translational potential as the results obtained might indicate that Lgr5⁺ liver stem cells might have the potential to alleviate liver fibrosis in chronic liver damage. I have only 2 comments:

1) in Fig 1b the authors mention that the GFP staining identified the transplanted Lgr5⁺ population. At what time point after transplantation is this done?

We thank the reviewer for pointing these out for us, the GFP staining were done on day 40, we added this information into the text and figure legends accordingly.

Similarly, what is not addressed is what cell types do these Lgr5⁺ cells contribute to after transplant. Do they only generate hepatocytes? Ductal cells? Other cells? In the absence of lineage tracing maybe the authors can speculate on their discussion?

We thank the reviewer for the valuable suggestions. We discussed the mechanism through which Lgr5⁺ cells prevent liver fibrosis in results section accordingly:

“However, because the lineage-tracing model is not currently available in our lab, it is not clear to which cell types do these Lgr5⁺ cells contribute to after transplan. According to an *in vitro* differentiation assay, transplanted Lgr5⁺ liver stem cells might primarily generate more hepatocytes in the host.”

2) Another question that arises from this manuscript, is the molecular mechanism by which the injection of these Lgr5⁺ cells results in a

reduction of liver fibrosis. Is the fibrosis cleared or simply not happening? While that could be a follow-up story out of the scope of this manuscript, it would be interesting for the authors to discuss their thoughts on that on the discussion section.

We thank the reviewer for the valuable suggestions. We discussed the mechanism through which Lgr5⁺ cells prevent liver fibrosis in discussion section:

“According to our current data, Lgr5⁺ liver stem cells induction not only inhibited the development of liver fibrosis but also attenuated established fibrosis. New healthy hepatocytes derived from Lgr5⁺ liver stem cells probably contributed to liver function recovery. Furthermore, our ongoing studies have demonstrated that Lgr5⁺ liver stem cells conditional medium can reverse activated hepatic stellate cells (HSCs) to quiescent HSCs, which indicates that Lgr5⁺ liver stem cells may reduce liver fibrosis through the direct secretion of specific factors. Further studies are needed to explore the detailed mechanisms.”

Point-by-point response (in blue) to Editor's concerns:

Editor's concerns:

Please use suggested text:

the authors declare no competing financial interests

We thank the editor for pointing this out for us. We edited the text according to editor's suggestion.

Please consider using the edited abstract. I have edited it so that it conforms with our journal style (eg max 150 words, use of present tense, needs to contain a “Here we show” phrase... see checklist for details)

Induction of endogenous adult stem cells by administering soluble molecules provide an advantaged approach for tissue damage repair, which could be a clinically applicable and cost-effective alternative to transplantation of embryonic or pluripotent stem cells-derived tissues for the treatment of acute organ failures.

Here we show that HGF and Rspo1 induce liver stem cells to rescue liver dysfunction.

Carbon tetrachloride treatment causes both fibrosis and Lgr5+ liver stem cell proliferation, whereas Lgr5 knockdown worsens fibrosis. Injection of HGF in combination with Rspo1 increases the number Lgr5+ liver stem cells and improves liver function by attenuating fibrosis. We find Lgr5+ liver stem cells in human liver fibrosis tissues, and once isolated these cells are able to form organoids and treatment with HGF/ Rspo1 promotes their expansion. We suggest that Lgr5+ liver stem cells represent a valuable target for liver damage treatment and

HGF/Rspo1 can be a potential therapy to promote liver stem cell expansion.

We thank the editor for editing the abstract for us. We edited abstract is perfect, we used this edited abstract in our revised manuscript now.

Please ensure subheadings in the results and methods section are no longer than 60 characters (incl punctuation) and do not contain punctuation (eg commas).

We thank the editor for pointing this out for us. We have edited all the subheadings in the results and methods section according to the editor's requirements.

Supplementary Fig.1

Please use this style when citing Supplementary Figures/Tables (ie, please don't use the prefix "S" to denote Supplementary material). Please change throughout the manuscript.

We thank the editor for pointing this out for us. We used the right style in our revised manuscript according to the editor's requirements.

We strongly discourage the use of "data not shown". Please show these data,

describe them numerically in the text or amend the statement. Please do the same for all other instances of "data not shown" in the text (if applicable).

We thank the editor for pointing this out for us. We showed the data as supplementary Fig.14c and d, and there is no "data not shown" in the revised manuscript.

Please explain the sequence of events, ie was liver tissue resected as a result of the accident? Or was it an emergency surgery and obtained consent retrospectively? Or was this surgery performed during recovery of these people in hospital after obtaining consent? Please also provide age and gender of the patients.

We thank the editor for pointing this out for us. We edited the Ethics Statement section and provide the information about age and gender of the patients as Supplementary Table 2, Table 3.

“The livers without fibrosis were harvested from 5 victims of car accidents with traumatic hepatorrhesis. Debridement of liver trauma was performed for these victims, and the liver tissues were resected for routine pathological biopsy; the samples used in our study were from these biopsy samples. The victims who donated their liver material were not deceased, and all of them provided informed consent retrospectively to use the material for this study. The livers with fibrosis were obtained from 42 patients with clinically diagnosed liver fibrosis, their liver tissues were harvested for routine pathological biopsy; the samples used in our study were from these biopsy samples. All of them provided informed consent retrospectively to use the material for this study. The patients information is provided in Supplementary Table 2 and Table 3.”

Please clarify the addition of this sentence here.

This refers to animal studies and not human studies.

We thank the editor for pointing this out for us. We edited the Ethics Statement section according to the editor's comments.

All study protocols were approved by the Clinical Research Ethics Committee of Southern Medical University and Wenzhou Medical University. Our policy is consistent with the principles embodied in the Declaration of Helsinki. All procedures involving animals, including housing and care, the method by which they were killed, and experimental protocols, were approved by the University Committee on Use and Care of Animals (UCUCA) of Southern Medical University. Our policy is consistent with the U.S. Public Health Service Policy on Use of Laboratory Animals, available from the Office of Laboratory Animal Welfare, National Institutes of Health.

Please state also the gender of the mice

We thank the editor for pointing this out for us. We used 8-week old male mice in our experiment, we edited the information accordingly in our revised manuscript.

Change to Immunofluorescence

Please state antibody concentrations/dilutions used, as well as catalogue and/or clone numbers (or their respective non-commercial sources) for all antibodies used in this study.

We thank the editor for pointing this out for us. We change "immunofluorescent" to immunofluorescence". And state antibody concentrations/dilutions used, as well as catalogue and/or clone numbers for all antibodies used in this study.

Please provide the gating strategy

We thank the editor for pointing this out for us. We provided the gating strategy in the revised manuscript.

“Single cell suspension was subjected for flow cytometric sorting and FL-1(AF488) channel was chosen to gate Lgr5-GFP⁺ liver stem cells.”

Please ensure that all necessary methodological details are provided in this manuscript so that it stands alone from other publications. Please do the same for all other instances where you use the term ‘as described previously’ or similar. We have recently abolished word limits for the methods section, so please provide describe all methods in sufficient detail.

I have noticed there are some methods missing (e.g. western blots)

We thank the editor for pointing this out for us. We provided all necessary methodological details and missing methods in the revised manuscript.

We and all the Nature journals have introduced data availability statements in the methods section of published papers. Please add a new subheading called “data availability” here and follow the instructions in the following document:

<http://www.nature.com/authors/policies/data/data-availability-statements-data-citations.pdf>

We thank the editor for pointing this out for us. We finished this work following the instructions.

Please delete it

We thank the editor for pointing this out for us. We deleted this part in the revised manuscript.

Please delete all the figures from the article file leaving only the figure legends. Figures need to be uploaded as separate files.

We thank the editor for pointing this out for us. We deleted all the figures from the article file leaving only the figure legends according to the editor's comments in the revised manuscript.

Please ensure that all microscopic images (or one per panel if resolution is the same) contain a scale bar and define it in the legend. Please apply to all figures containing microscopic images.

We thank the editor for pointing this out for us. We corrected all the images according to the editor's comments in the revised manuscript.

Please define error bars and state name of statistical tests used to generate p values. Please make sure you do the same for all other figures, including those in the supplementary information.

We thank the editor for pointing this out for us. We modified them according to the editor's comments in the revised manuscript.

Please ensure that all western blots and gels are accompanied by the locations of molecular weight/size markers. Blots should be cropped such that at least one marker position is present. Please also supply uncropped scans of the most important western blots as a supplementary figure in the supplementary information and cite the new supplementary figures in the text.

We thank the editor for pointing this out for us. We modified them according to the editor's comments in the revised manuscript. We provided uncropped scans of the most important western blots as supplementary fig. 8).

Has any of the images shown in the figures or supplementary figures been published before, or have they been adapted from previously published images? We strongly discourage the use or adaptation of previously published images, but if this is unavoidable, please request the necessary rights documentation to re-use such material from the relevant copyright.

No, the images were designed originally by ourselves.